# Recombination occurs within minutes of replication blockage by *RTS1* producing restarted forks that are prone to collapse

Michael O Nguyen, Manisha Jalan, Carl A Morrow, Fekret Osman, Matthew C Whitby*

Department of Biochemistry, University of Oxford, Oxford, United Kingdom

**Abstract** The completion of genome duplication during the cell cycle is threatened by the presence of replication fork barriers (RFBs). Following collision with a RFB, replication proteins can dissociate from the stalled fork (fork collapse) rendering it incapable of further DNA synthesis unless recombination intervenes to restart replication. We use time-lapse microscopy and genetic assays to show that recombination is initiated within ~10 min of replication fork blockage at a site-specific barrier in fission yeast, leading to a restarted fork within ~60 min, which is only prevented/curtailed by the arrival of the opposing replication fork. The restarted fork is susceptible to further collapse causing hyper-recombination downstream of the barrier. Surprisingly, in our system fork restart is unnecessary for maintaining cell viability. Seemingly, the risk of failing to complete replication prior to mitosis is sufficient to warrant the induction of recombination even though it can cause deleterious genetic change.

## Introduction

The completion of eukaryotic DNA replication relies on the merging of replication forks that emanate from multiple origins distributed along each chromosome. The failure of even one pair of replication forks to merge results in a region of unreplicated DNA, which can lead to chromosome missegregation and/or DNA breakage during mitosis, and ultimately genomic changes that can drive the development of diseases such as cancer.

The progression of replication forks, from origin to point of merging, is frequently hindered by obstacles in and on the DNA template, including various DNA lesions and protein–DNA complexes (*Lambert and Carr, 2013*). In many cases, these cause only a transient stalling of replication during which the replication proteins (replisome) remain engaged with the DNA by a fork protection mechanism mediated by the intra-S-phase checkpoint (*Errico and Costanzo, 2012*). However, the replisome can sometimes dissociate to produce a collapsed fork, which may also be associated with breakage of the DNA, and this is especially prevalent following oncogene activation in human cells (*Halazonetis et al., 2008*). Homologous recombination (HR) can restore fork integrity and restart replication, via a process termed break-induced replication (BIR) (*Anand et al., 2013*; *Costantino et al., 2014*). BIR has mainly been studied in the budding yeast *Saccharomyces cerevisiae* away from the context of DNA replication, using systems that generate a site-specific DNA double-strand break (DSB) that is repaired by recombination with a donor DNA molecule containing sequence homology to only one side of the break (*Anand et al., 2013*). It is, therefore, unclear to what extent these studies reflect how recombination acts to repair stalled and broken replication forks during S-phase.

In BIR, the DSB is resected to generate a single-stranded DNA (ssDNA) tail, which is bound firstly by Replication Protein A (RPA) and then Rad52. Rad52 mediates the subsequent loading of Rad51, which catalyses the key steps of homologous DNA pairing and strand invasion to form a displacement

*For correspondence: matthew. whitby@bioch.ox.ac.uk

Competing interests: The authors declare that no competing interests exist.

**eLife digest** Before a cell can divide to form two new cells, it must duplicate its DNA to ensure the newly formed cells have the same genetic information as the original. This genetic material is made up of two single strands of DNA that are paired to form a double-stranded helix. Certain groups of proteins separate these two DNA strands to form a two-pronged structure known as a 'replication fork'. This occurs at different points along the length of the DNA double helix. Groups of proteins then travel down the DNA strands, separating them as they go, and using them as templates for making copies of the DNA.

DNA replication is finally completed when different replication forks meet and merge. This process does not always occur smoothly because some regions of DNA contain obstacles that impede the movement of the replication machinery. In most cases, the replication proteins briefly stall and then restart. However, occasionally the machinery can fall off the DNA; this event is known as a 'replication fork collapse'.

Nguyen et al. have now used a method called time-lapse microscopy to visualise this process inside a species of yeast—called fission yeast—as it occurs in real time. Fission yeast's genetic material is known to contain a specific site that blocks the replication machinery. Nguyen et al. found that a protein called Rad52 arrives at this specific site within 10 minutes of a replication fork being blocked. This protein enables recovery of the replication fork within an hour via a process known as 'DNA recombination'.

Nguyen et al. also unexpectedly found that the restarted fork is susceptible to further collapse. It is known from previous work that the mechanisms that repair broken DNA and rescue replication forks can also introduce errors into the DNA. This implies that if fork collapse occurs frequently, it can lead to the introduction of numerous errors that can be detrimental to cells. Thus, the rescue of fork collapse is like a double-edged sword; it is required for replication to proceed, but can lead to genetic changes inside cells. The failure to faithfully replicate genetic material drives the development of diseases such as cancer. Therefore, insights gained from Nguyen et al.'s findings may provide an improved understanding of how genetic alterations occur in both normal and cancerous cells.

(D) loop. The 3' end of the invading DNA strand primes DNA synthesis by polymerase δ, and replication then proceeds in a conservative manner involving migration of the D-loop, dependent on the Pif1 DNA helicase (*Saini et al., 2013*; *Wilson et al., 2013*). Similar to other modes of DSB repair by HR, the initial steps of BIR from DSB resection to strand invasion occur within ~30 min of DSB formation (*Jain et al., 2009*; *Hicks et al., 2011*). In contrast, the transition from strand invasion to DNA replication is delayed for several hours by a checkpoint that senses whether both ends of the DSB can engage with the same donor DNA sequence in a manner that is productive for completing repair (*Malkova et al., 2005*; *Jain et al., 2009*). However, once initiated DNA synthesis proceeds at a rate of 3–4 kb/min, which is comparable with normal DNA replication (*Malkova et al., 2005*).

Unlike normal DNA replication, BIR is highly error-prone with greatly increased rates of polymerase errors that remain uncorrected (*Deem et al., 2011*). There are also frequent dissociations of the elongating strand from the D-loop within a 10 kb window downstream of the DSB (*Smith et al., 2007*; *Stafa et al., 2014*), which results in multiple rounds of strand re-invasion increasing the risk of recombination between ectopic homologous DNA sequences that can give rise to gross chromosome rearrangements and copy-number variations. This feature of BIR is thought to reflect an inherent drive to repair the DSB by synthesis-dependent strand annealing (SDSA) (*Smith et al., 2007*). However, beyond 10 kb the D-loop appears to be stabilized, and BIR proceeds without the continual interruption of D-loop dissociation.

Whilst it is clear that a broken replication fork requires HR for DNA repair, it is less certain that it would be needed at a collapsed, yet unbroken, fork where replication could be completed most simply by convergence with the opposing fork. However, replication fork collapse at a site-specific protein–DNA fork barrier called *RTS1*, in the fission yeast *Schizosaccharomyces pombe*, has been shown to provoke DSB-independent recombination-dependent replication (RDR), which is required for viability (*Lambert et al., 2005*, *2010*). Similar to BIR, *RTS1*-induced RDR is error-prone, at least

within the first 2.4 kb downstream of the barrier (*Iraqui et al., 2012*; *Mizuno et al., 2013*). However, it is unknown whether it suffers the same tendency for D-loop dissociation as BIR. It also remains unclear whether recombination is a default response to replication fork blockage at *RTS1*, how quickly it initiates, and whether it only plays a role in restart at the blocked fork, or is also needed to promote fork merging.

To address these questions, we have used time-lapse microscopy to obtain single-cell resolution of RDR in response to replication fork blockage at *RTS1*, in combination with genetic assays to assess restarted fork fidelity. We show that Rad52 is recruited to *RTS1* in the majority of cells within minutes of fork blockage and seemingly gives rise to RDR without the long delay in progressing from strand invasion to DNA synthesis that is characteristic of BIR. However, like BIR the restarted fork is prone to multiple rounds of strand disengagement and reengagement, implying that even at a blocked replication fork during S-phase the default response is to attempt SDSA. Unlike BIR, this continues for more than 10 kb downstream of the barrier with little sign of abating, giving rise to a dramatic increase in recombination in this region. We also find that fork convergence, rather than inducing recombination, acts to prevent or curtail it. Surprisingly, despite the high frequency of the recombination response and contrary to previous reports, we find no evidence that it is required for cell viability. Seemingly, the risk of failing to complete DNA replication in a timely fashion is sufficient to warrant the initiation of RDR as a default response to replication fork collapse, even though in many cases it may be unnecessary and even cause deleterious genetic change.

## Results

### Experimental system

We have previously shown that replication fork blockage at the replication terminator sequence *RTS1* is sufficient to induce HR between a direct repeat of *ade6*⁻ heteroalleles in *S. pombe* (*Ahn et al., 2005*) (*Figure 1A*). *RTS1* is a unidirectional replication fork barrier (RFB), which consists of cis-acting DNA sequence elements and trans-acting factors including the myb domain-containing protein Rtf1 (*Codlin and Dalgaard, 2003*; *Eydmann et al., 2008*). Replication of the *ade6* locus is essentially unidirectional due to the relative position of replication origins that flank it (*Figure 1A*). Consequently, only one orientation of *RTS1* causes replication fork blockage at this site, which we will refer to as the active orientation (AO). The opposite orientation, which does not block replication, will be referred to as the inactive orientation (IO). Native two dimensional (2D) gel electrophoresis analysis of replication intermediates in an EcoNI fragment containing *RTS1* confirms that *RTS1-AO* strongly blocks replication forks, whereas *RTS1-IO* does not (*Figure 1B*). It also shows that a proportion of forks remain blocked at *RTS1-AO* long enough for replication to be completed by the opposing fork resulting in fork merging at *RTS1* as indicated by the appearance of double Y-shaped molecules (*Figure 1B*, *Figure 1—figure supplement 1A,B*). The remaining blocked forks appear to be able to restart and replicate past the barrier as indicated by the presence of large Y-shaped DNA molecules (*Figure 1B*, *Figure 1—figure supplement 1A,B*). The ratio of large Ys to double Ys is approximately 1:1 suggesting that about half the forks blocked at *RTS1-AO* restart prior to fork convergence (*Figure 1—figure supplement 1B*).

### Rad52 is required for replication past *RTS1-AO*

Work from the Carr and Lambert laboratories has shown that replication restart from *RTS1* depends on HR (*Lambert et al., 2010*), and consistent with this we observe a 99-fold increase in gene conversions and a 33-fold increase in deletions between the *ade6*⁻ heteroalleles that flank *RTS1-AO* compared to background levels observed without *RTS1* or with *RTS1-IO* (*Figure 1C*, *Table 1*) (*Ahn et al., 2005*). Moreover, this elevated frequency of HR is totally dependent on Rad52 (*Figure 1C*, *Table 1*), which is essential for both RFB-induced RDR and BIR in yeast (*Malkova et al., 1996*; *Lambert et al., 2010*).

Evidence that Rad52 restarts replication at *RTS1* (positioned at the *ura4* locus and under inducible Rtf1 control) includes the observation that there is more fork convergence at the barrier in a *rad52Δ* mutant than in wild type, as judged by the accumulation of double Y-shaped DNA molecules on 2D gels (*Lambert et al., 2010*). However, in previous work from our laboratory, we failed to detect such an increase at *RTS1-AO* suggesting that RDR does not occur at appreciable levels in our experimental system (*Lorenz et al., 2009*). The method of DNA extraction for 2D gel analysis can have a significant

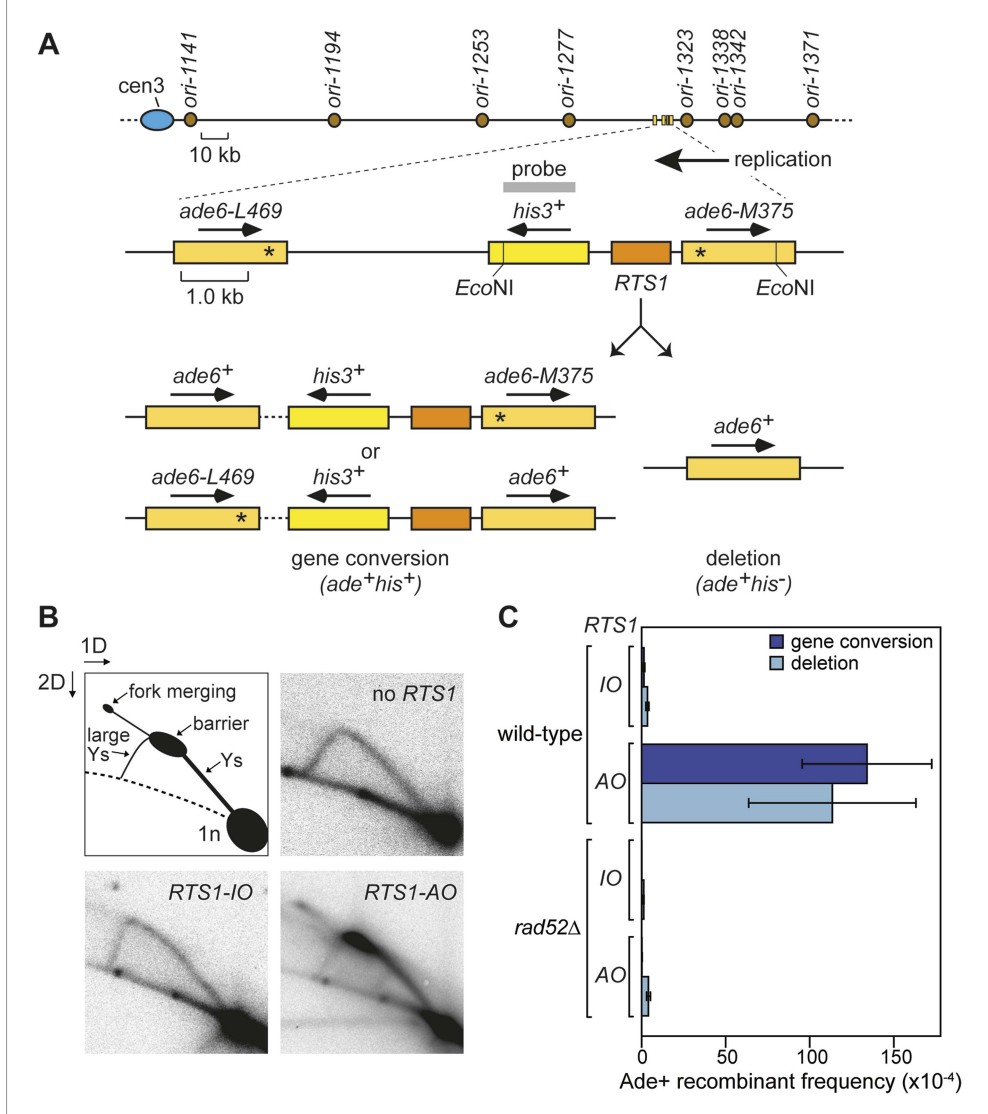

**Figure 1**. Experimental system for studying RFB-induced RDR. (**A**) Schematic showing the location of the direct repeat recombination reporter on chromosome 3, the two types of recombinants, and the position of the probe used for 2D gel analysis in **B**. Asterisks indicate the position of point mutations in *ade6-L469* and *ade6-M375*. (**B**) 2D gel analysis of replication intermediates in the EcoNI fragment shown in **A**. The DNA was extracted from strains MCW429 (no *RTS1*), MCW4712 (*RTS1-IO*), and MCW4713 (*RTS1-AO*). (**C**) Ade[+] recombinant frequencies for strains MCW4712, MCW4713, MCW1687, and MCW1688. Data are represented as mean ± SD.

The following figure supplement is available for figure 1:

**Figure supplement 1**. The proportion of restarted forks to converged forks at *RTS1-AO*.

effect on the ability to detect different types of replication intermediates (*Liberi et al., 2006*). Our standard protocol is based on mechanical cell disruption to extract nuclei, DNA purification by caesium chloride density gradient centrifugation, and enrichment for replication intermediates on benzoylated napthoylated DEAE (BND) cellulose columns (*Lorenz et al., 2009*). In contrast, *Lambert et al. (2010)* enzymatically lyse cells embedded in agarose plugs prior to enrichment of replication intermediates on BND cellulose. To determine whether these differences could account for our failure to detect an increase in converging forks, we compared replication intermediates in the EcoNI fragment containing *RTS1-AO* from wild-type and a *rad51Δ rad52Δ* double mutant using the same

**Table 1**. Direct repeat recombinant frequencies

| Genotype | RTS1 orientation | Position of direct repeat | Colonies analysed | Ade$^+$ His$^+$ recombinant frequency ($\times 10^{-4}$)* | | Ade$^+$ His$^-$ recombinant frequency ($\times 10^{-4}$)* | |
|---|---|---|---|---|---|---|---|
| | | | | Mean | p value† | Mean | p value† |
| wild type | IO | Flanking RTS1 | 77 | 1.36 (+/− 0.51) | – | 3.48 (+/− 0.89) | – |
| wild type | AO | Flanking RTS1 | 77 | 134.03 (+/− 38.50) | – | 113.41 (+/− 49.72) | – |
| rad52Δ | IO | Flanking RTS1 | 15 | 0.03 (+/− 0.04) | <0.001‡ | 1.20 (+/− 0.21) | <0.001‡ |
| rad52Δ | AO | Flanking RTS1 | 15 | 0.13 (+/− 0.11) | <0.001§ | 4.07 (+/− 1.16) | <0.001§ |
| ori-1253Δ | IO | Flanking RTS1 | 15 | 2.33 (+/− 1.86) | 0.060‡ | 4.80 (+/− 1.58) | 0.007‡ |
| ori-1253Δ | AO | Flanking RTS1 | 16 | 339.88 (+/− 83.61) | <0.001§ | 221.02 (+/− 57.21) | <0.001§ |
| wild type | IO | Site A | 18 | 1.44 (+/− 0.86) | 0.698‡ | 4.61 (+/− 1.66) | 0.011‡ |
| wild type | AO | Site A | 21 | 128.66 (+/− 43.85) | 0.746§ | 680.48 (+/− 305.44) | <0.001§ |
| wild type | IO | Site B | 18 | 0.88 (+/− 0.51) | 0.023# | 2.45 (+/− 1.88) | 0.001# |
| wild type | AO | Site B | 15 | 5.12 (+/− 2.41) | <0.001¶ | 85.02 (+/− 33.72) | <0.001¶ |
| ori-1253Δ | IO | Site B | 15 | 1.58 (+/− 0.50) | <0.001** | 4.86 (+/− 2.56) | 0.006** |
| ori-1253Δ | AO | Site B | 15 | 86.03 (+/− 33.63) | <0.001†† | 1094.46 (+/− 443.32) | <0.001†† |

*The values in parentheses are the standard deviations about the mean.
†p values are derived from independent-sample t-tests comparing the mean values as indicated.
‡Compared to the equivalent mean recombinant frequency in wild type with RTS1-IO flanked by ade6⁻ direct repeats.
§Compared to the equivalent mean recombinant frequency in wild type with RTS1-AO flanked by ade6⁻ direct repeats.
#Compared to the equivalent mean recombinant frequency in wild type with RTS1-IO and site A ade6⁻ direct repeats.
¶Compared to the equivalent mean recombinant frequency in wild type with RTS1-IO and site A ade6⁻ direct repeats.
**Compared to the equivalent mean recombinant frequency in wild type with RTS1-IO and site B ade6⁻ direct repeats.
††Compared to the equivalent mean recombinant frequency in wild type with RTS1-AO and site B ade6⁻ direct repeats.

protocol of DNA extraction as *Lambert et al. (2010)*. A *rad51Δ rad52Δ* double mutant was used instead of a *rad52Δ* single mutant because of the latter's susceptibility to acquire suppressor mutations that enable Rad51 to catalyse HR in the absence of Rad52 (*Osman et al., 2005*). Unlike *Lambert et al. (2010)*, we did not observe an increase in double Y-shaped molecules, in fact they decreased by ~twofold in the *rad51Δ rad52Δ* mutant (*Figure 2A,B*). However, the reduction in large Ys was even greater (~sixfold) (*Figure 2B*), such that their ratio to double Ys decreased more than threefold (*Figure 2C*). These data indicate that Rad52 plays an important role in promoting replication past *RTS1-AO*. We suspect that a *rad51Δ rad52Δ* mutant fails to manifest an increase in double Y-shaped molecules because the effect of an increased frequency of fork convergence at *RTS1-AO* is offset by a faster rate of fork merging, due to the absence of recombination proteins that could impede this process.

## Live cell imaging of RDR at *RTS1*

Whilst replication fork blockage at *RTS1* clearly induces recombination, it is unclear whether this happens in all cells or only a subset in each cell cycle. To address this question, we inserted an array of *lacO* sequences downstream of *RTS1* so that we could track its location in cells expressing the LacI repressor fused to the far-red fluorescent protein tdKatushka2 using time-lapse microscopy (*Figure 3A,B*). Mindful that the *lacO*-LacI interaction can act as a RFB (*Sofueva et al., 2011*), we determined imaging parameters under which the amount of LacI, whilst sufficient to detect the location of the *lacO* array through several hours of live cell imaging, was insufficient to perturb replication forks as judged by native 2D gel analysis (*Figure 3—figure supplement 1*) or induce a recombinational response (*Figure 3—figure supplement 2*). Having established suitable conditions, we imaged asynchronously growing cells containing either *RTS1-IO* or *RTS1-AO* and Rad52 fused to yellow fluorescent protein (YFP), which forms foci in response to DNA damage similar to its homologue in *S. cerevisiae* (*Lisby et al., 2001*, *2003*; *Meister et al., 2003*) (*Figure 3B*). Time-lapse

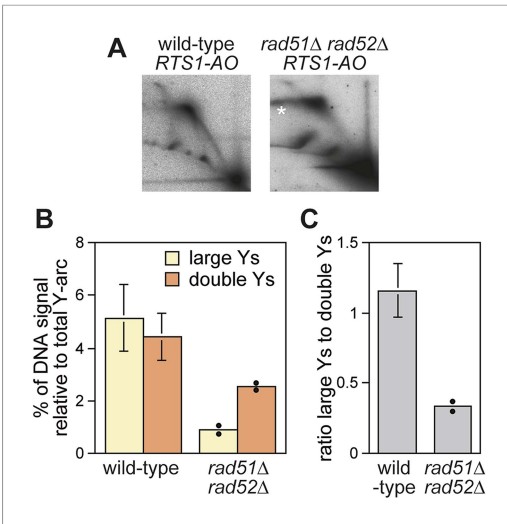

**Figure 2**. Recombination proteins are needed for replication past *RTS1-AO*. (**A**) 2D gel analysis of replication intermediates in the EcoNI fragment shown in *Figure 1A*. The DNA was extracted from strains MCW4713 (wild-type *RTS1-AO*) and MCW1696 (*rad51Δ rad52Δ RTS1-AO*) using the same method as in *Lambert et al. (2010)*. Note that the signal extending from the RFB signal indicated by the asterisk is an aberration of how the gel ran in the first dimension. (**B** and **C**) Quantification of 2D gels. Mean values (± SD) from four independent experiments for wild type are shown. In the case of *rad51Δ rad52Δ*, the values are based on two independent experiments with each value represented by a dot around the mean.

movies of cells, with images taken at 5-min intervals, were staged relative to anaphase, and the presence of Rad52 foci and their co-localization with *lacO*-LacI foci recorded over a 90-min period (*Figure 3C*). The majority (≥79%) of cells exhibited a single Rad52 focus at varying time points mainly between 20 and 90 min post-anaphase (*Figure 3C*). This is true even for cells without *RTS1* and reflects the fact that HR is routinely needed to process perturbed replication forks and DNA damage that occurs during S-phase (data not shown). Consistent with replication fork blockage at *RTS1* inducing HR, a higher percentage of cells with *RTS1-AO* exhibited a Rad52 focus than those with *RTS1-IO* (89% vs 79%), and on average, these foci were detected over a greater number of time points (*Figure 3C*, *Figure 3—figure supplement 3A,B*). Imaging of *RTS-AO* and *RTS-IO* cells without *lacO*-LacI revealed no difference in the occurrence and timing of Rad52 foci confirming that *lacO*-LacI acts as an inert marker for the location of *RTS1* (*Figure 3—figure supplement 3A,B*).

Analysis of the co-localization of Rad52 foci with *lacO*-LacI foci revealed a striking difference between cells with *RTS1-IO* and *RTS1-AO* (*Figure 3C*). ~20% of cells with *RTS1-IO* exhibited co-localizing foci, which on average were detectable in 1.5 time points, whereas with *RTS1-AO* this increased to ~60% of cells and 3.8 time points (*Figure 3C*). Cells without *RTS1* exhibit the same relatively low incidence of Rad52 focus co-localization with *lacO*-LacI as those containing *RTS1-IO* consistent with the observation that *RTS1-IO* does not induce recombination (data not shown). Together, these data show that replication fork blockage induces a recombinational response at *RTS1* in most but not all cells. Additionally, they show that blockage of a single replication fork is sufficient to induce a Rad52 focus. However, the overall increase in Rad52 foci observed in *RTS1-AO* cells is less than the number of co-localizing foci implying that some Rad52 foci that are engaged at *RTS1* can at the same time be engaged with other lesions/perturbed replication forks consistent with them acting as repair centres (*Lisby et al., 2003*).

## Both Rad51 and Rad54 are recruited to *RTS1-AO*

To determine whether Rad52 foci co-localizing with *lacO*-LacI represent sites of active recombination, rather than simply Rad52 binding to ssDNA at the stalled replication fork, we looked for the appearance of both Rad51 and Rad54 foci. In snapshots of asynchronously growing cells containing *RTS1-IO*, Rad52-YFP and Rad51 tagged at its N-terminus with cyan fluorescent protein (CFP), 8.1% of cells contain a Rad52 focus and 6.6% a Rad51 focus, with 95% of the latter co-localizing with a Rad52 focus (*Figure 4B*). Only 2.6% of Rad52 foci and 3.2% of Rad51 foci were observed to co-localize with *lacO*-LacI in these cells (*Figure 4B*). In cells with *RTS1-AO*, the overall percentage of cells with Rad52 and Rad51 foci increases to 13% and 12%, respectively, with the vast majority (97%) of Rad51 foci again co-localizing with a Rad52 focus (*Figure 4B*). Importantly, 36.5% of Rad52 foci and 37.9% of Rad51 foci co-localized with *lacO*-LacI, with greater than 90% of these foci also co-localizing with each other (*Figure 4A,B*). Cells containing Rad54 fused to green fluorescent protein (GFP), but without Rad52-YFP, were also imaged (*Figure 4C*). 11.7% of *RTS1-IO* cells contained a Rad54-GFP focus, but only 5.5% of these foci co-localized with *lacO*-LacI (*Figure 4D*). Similar to Rad51 and Rad52, the frequency of Rad54-GFP foci increased in *RTS1-AO* cells (from 11.7% to 14%) and a much greater

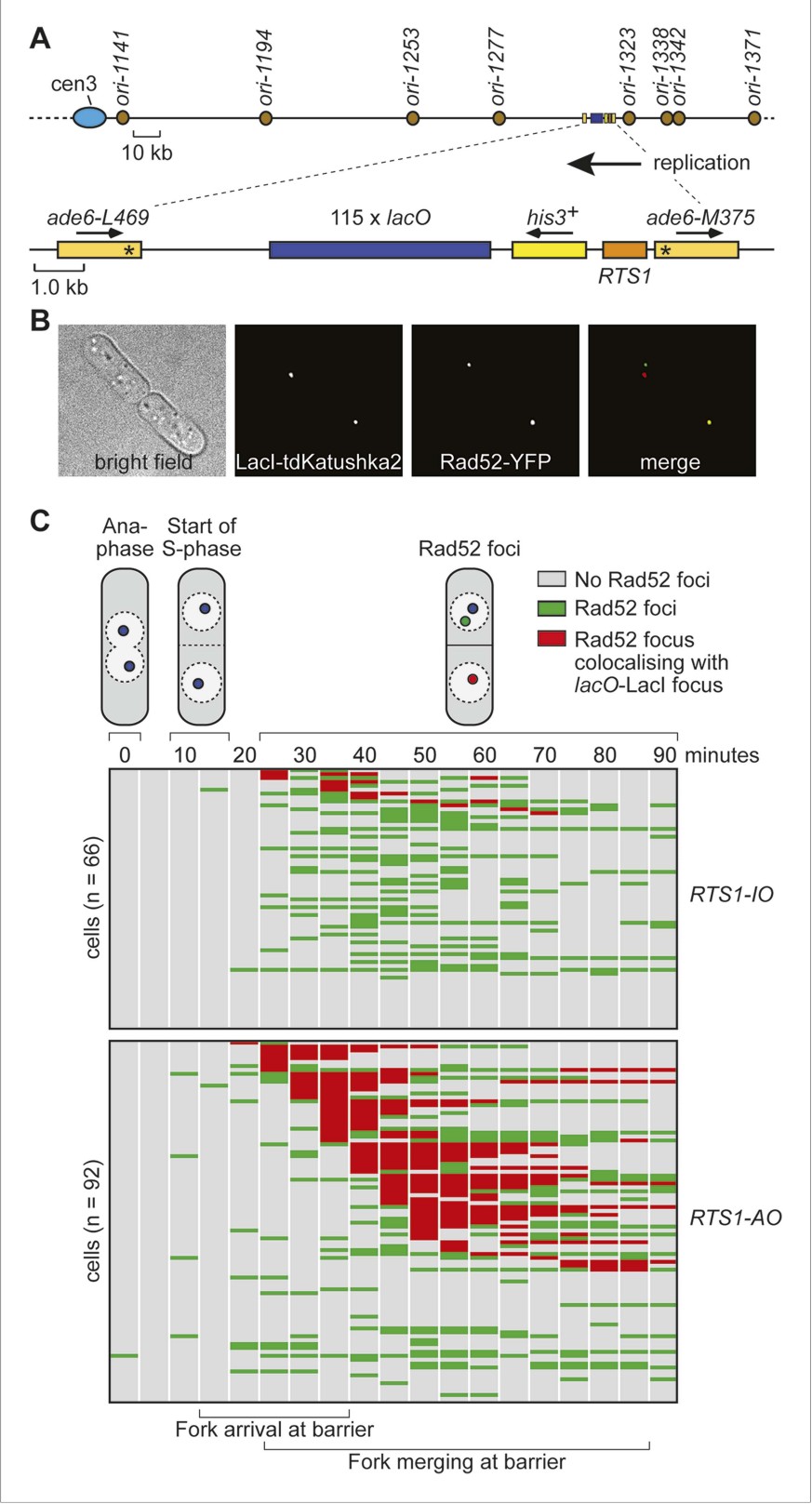

**Figure 3**. Tracking Rad52-YFP focus localization at *RTS1* by time-lapse microscopy. (**A**) Schematic showing the modification of the direct repeat recombination reporter for time-lapse microscopy. (**B**) Representative stills taken from a time-lapse movie of two daughter cells each with a LacI-tdKatushka2 and Rad52-YFP focus. In the right hand
*Figure 3. continued on next page*

*Figure 3. Continued*

daughter, the foci co-localize. (**C**) Analysis of time-lapse movies. The presence of a Rad52 focus and whether it co-localizes with the *lacO*-LacI-tdKatushka2 focus is recorded every 5 min for 90 min post-anaphase in each cell. The top panel is a schematic of *S. pombe* cells at various stages post-anaphase with nuclear *lacO*-LacI foci in blue, Rad52 foci in green, and co-localizing foci in red. The strains are MCW6395 and MCW6556.

The following figure supplements are available for figure 3:

**Figure supplement 1**. Effect of LacI expression on replication fork progression through the *lacO* array downstream of *RTS1-AO*.

**Figure supplement 2**. Effect of LacI expression on the frequency of direct-repeat recombination in strains containing a *lacO* array between the repeats.

**Figure supplement 3**. Frequency of Rad52 foci in *RTS1-IO* and *RTS1-AO* cells.

percentage of these foci (36.1%) co-localized with *lacO*-LacI (*Figure 4C,D*). Collectively, these data indicate that both Rad51 and Rad54 are recruited to *RTS1-AO* to a similar extent as Rad52. Moreover, the coincidence of Rad51 and Rad52 foci suggests that these proteins are functioning together at the *RTS1* barrier, and with Rad54 are likely to represent active and ongoing recombination.

## Timing of Rad52 recruitment following replication fork blockage

To gauge how quickly Rad52 is recruited to forks blocked at *RTS1-AO*, we first assessed the timing of S-phase relative to anaphase by imaging cells expressing the replication marker PCNA fused to CFP (*Meister et al., 2005*). CFP-PCNA forms patterns of foci that are characteristic for the different stages of S-phase (*Meister et al., 2007*) (*Figure 5—figure supplement 1*), and under our experimental conditions these foci first appear between 10 and 15 min post-anaphase, marking the start of S-phase (*Figure 5A*). In cells containing *RTS1-AO*, Rad52 foci co-localizing with *lacO*-LacI first appear between 20 and 25 min post-anaphase (*Figures 3C and 5B*). This lag between the start of S-phase and appearance of Rad52 foci at the RFB is also observed when both CFP-PCNA and Rad52-YFP are imaged concurrently in the same cells (*Figure 5C*). As the nearest replication origin to *RTS1-AO* is ~7.7 kb away (*Figure 3A*, *Table 2*), the earliest a fork can reach the barrier is ~2.6 min after the start of S-phase, based on an average fork velocity of ~3 kb/min (*Heichinger et al., 2006*). Therefore, Rad52 can be recruited to *RTS1-AO* as early as 7.4–12.4 min after replication fork blockage. The later appearance of Rad52 foci at *RTS1-AO* that is observed in many cells (ranging mainly from 30 to 55 min post-anaphase) (*Figure 3C*), likely reflects the later blockage of forks that emanate from more distal origins (i.e., *ori-1338*, *ori-1342*, and *ori-1371*), which would arrive at the barrier from 7.7 to 18.5 min after the start of S-phase (*Figure 3A,C*, *Table 2*). In a few cells (~5%), we observed Rad52 foci first co-localizing with *lacO*-LacI at 60–80 min post-anaphase (*Figure 3C*). These very late appearing co-localizing foci may represent cells in which one or both forks converging on *RTS1* have been delayed by other RFBs or possibly a recombinational response to occasional problems that might occur during fork convergence.

## Duration of Rad52 at the blocked replication fork

As mentioned above, Rad52 foci remain co-localized with *lacO*-LacI in cells with *RTS1-AO* for an average of 3.8 time points. If one considers only the longest track of consecutive time points with a co-localization in each cell, then the duration of any single-event ranges from 1 to 9 time points (average = 3.4 time points), which equates to >0 to <45 min (*Figure 3C*). However, as these estimates are based on images taken every 5 min, we cannot be certain that Rad52 does not dissociate and then re-associate with the site in between points of image acquisition. The percentage of cells with PCNA foci begins to decline approximately 40–45 min after anaphase (i.e., 30–35 min after the start of S-phase) and by 70 min they are detectable in less than 20% of all cells (*Figure 5A*). S-phase in *S. pombe* has been calculated to take ~20 min (*Mitchison and Creanor, 1971*; *Heichinger et al., 2006*); however, the persistence of PCNA foci suggests that the

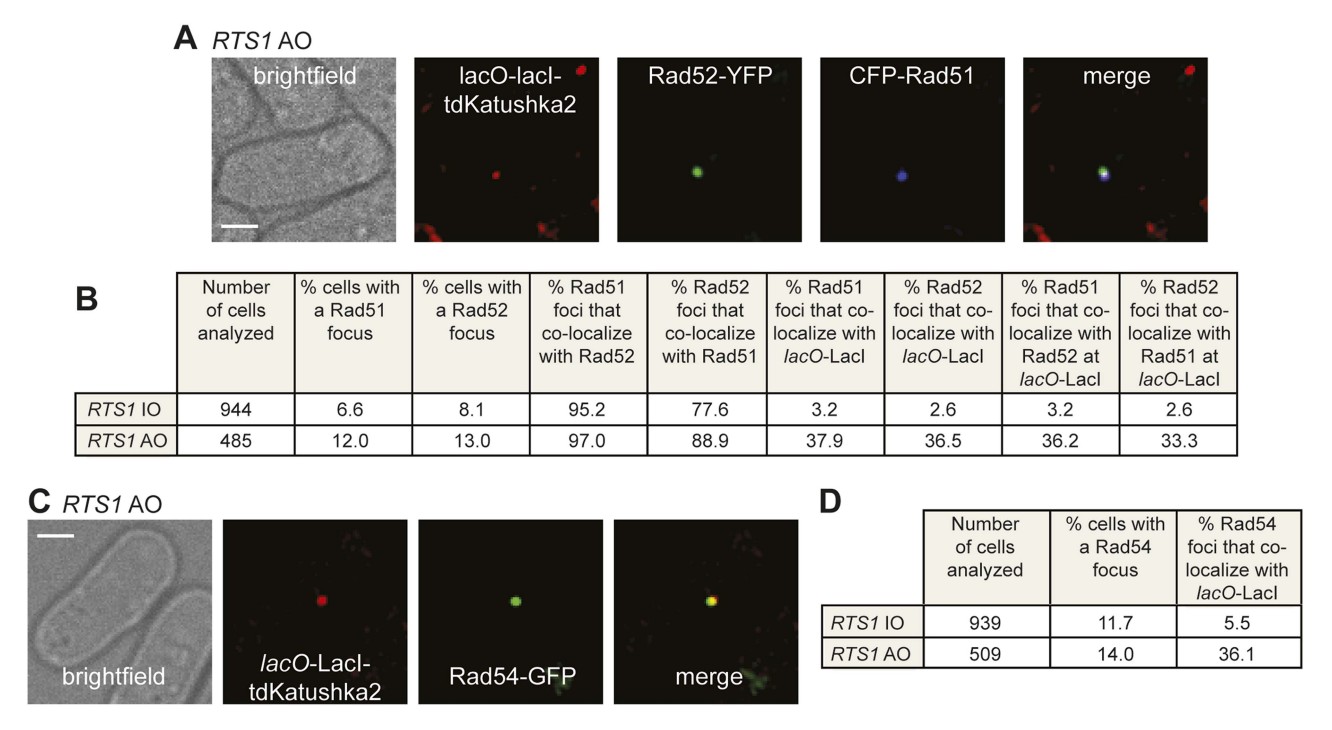

Figure 4. Both CFP-Rad51 and Rad54-GFP foci co-localize with *lacO*-LacI in *RTS1-AO* cells to a similar extent as Rad52-YFP foci. (**A**) Example snapshots of a *RTS1-AO* cell with both a CFP-Rad51 and Rad52-YFP focus co-localizing with a *lacO*-LacI-tdKatushka2 focus. The scale bar represents 2 μm. (**B**) Quantification of data like in **A**. The strains are MCW7640 (*RTS1-IO*) and MCW7638 (*RTS1-AO*). (**C**) Example snapshots of a *RTS1-AO* cell with a Rad54-GFP focus co-localizing with a *lacO*-LacI-tdKatushka2 focus. The scale bar represents 2 μm. (**D**) Quantification of data like in **C**. The strains are MCW7646 (*RTS1-IO*) and MCW7645 (*RTS1-AO*).

completion of DNA replication can take a lot longer and will often span into what is considered to be G2-phase. By comparing the timings of when Rad52 co-localizes with *lacO*-LacI with those when PCNA foci are detectable (*Figure 5A,B*), we conclude that in most cells Rad52's association with *RTS1-AO* ends before the disappearance of PCNA foci, that is, during S-phase or early G2-phase. Indeed, when Rad52-YFP and CFP-PCNA are imaged simultaneously, the co-localization of Rad52 with *lacO*-LacI terminates before the disappearance of PCNA foci in ~71% of cells with *RTS1-AO* (*Figure 5D*). The remaining ~29%, which exhibit a *lacO*-LacI co-localizing Rad52 focus beyond the point at which PCNA foci are no longer detectable, may represent cells with ongoing RDR in G2-phase or where fork convergence at *RTS1* has been problematic and therefore caused a recombinational response.

In a minority of cells (~15%) with *RTS1-AO*, Rad52 focus co-localization with *lacO*-LacI is discontinuous, with gaps of up to 50 min between points of co-localization (*Figure 3C*). Although the significance of this remains uncertain, it does suggest that replication fork blockage by *RTS1* can give rise to distinct and temporally separable problems in the vicinity of the *lacO* array that in each case provoke recombination.

## The recruitment and duration of Rad52 at *RTS1-AO* is influenced by the stochastic nature of origin firing

Eight main replication origins flank *RTS1-AO*, and each fires in early S-phase with efficiencies ranging from 22% to 84% (*Heichinger et al., 2006*; *Daigaku et al., 2015*). Based on these data, the known location of the origins and a fork velocity of 3 kb/min, we can conclude that the time between replication fork blockage and convergence at *RTS1* will vary from cell-to-cell, and we have made an estimate for what these times will be (*Figure 6A*). In some cells, fork convergence will occur within 10 min of the first fork arriving at *RTS1-AO*, providing insufficient time for Rad52 recruitment before

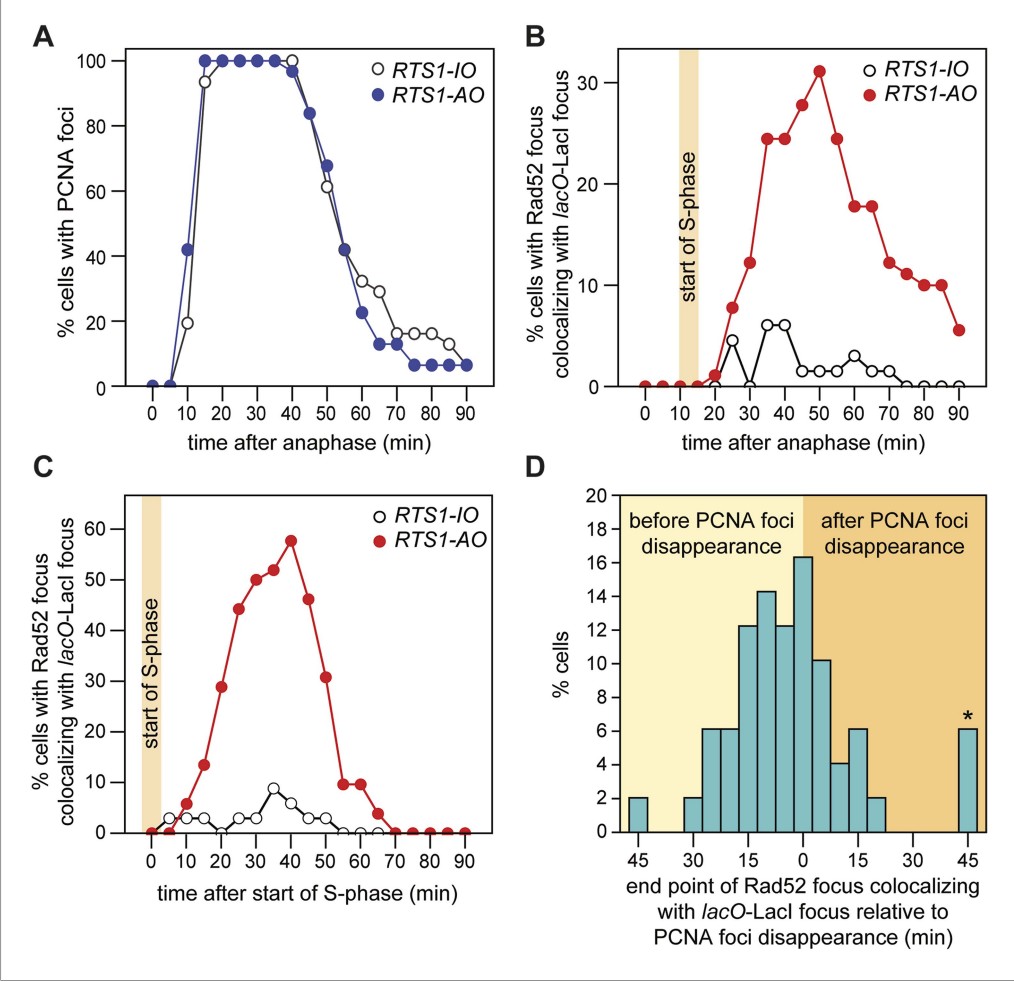

**Figure 5**. Temporal kinetics of Rad52-YFP foci localization to *RTS1*. (**A**) Percentage of cells with one or more CFP-PCNA foci in the first 90 min post-anaphase. The *RTS1-IO* strain is MCW6701 (n = 31), and the *RTS1-AO* strain is MCW6706 (n = 31). (**B**) Percentage of cells with a Rad52-YFP focus that co-localizes with the *lacO*-LacI focus in the first 90 min post-anaphase. The data are derived from **Figure 3C**. (**C**) Percentage of cells with a Rad52-YFP focus that co-localizes with the *lacO*-LacI focus in the 90 min following the first appearance of CFP-PCNA foci post-anaphase (i.e., the start of S-phase). The *RTS1-IO* strain is MCW6712 (n = 34) and the *RTS1-AO* strain is MCW7065 (n = 52). Note that the overall percentage of cells with a Rad52-YFP focus co-localizing with *lacO*-LacI is slightly higher than in cells without CFP-PCNA. The CFP tag on PCNA partially impairs its function, which is compensated by expression of untagged PCNA in the same cells (**Meister et al., 2007**). Nevertheless, we suspect that a subtle impairment of PCNA function accounts for the more frequent co-localization of Rad52-YFP with *lacO*-LacI. (**D**) End point of Rad52-YFP focus co-localization with the *lacO*-LacI focus relative to the disappearance of CFP-PCNA foci. The strain is MCW7065 (n = 49). The asterisk indicates cells with a Rad52-YFP focus co-localizing with the *lacO*-LacI focus more than 45 min after CFP-PCNA foci disappearance.

The following figure supplement is available for figure 5:

**Figure supplement 1**. Representative stills from a time-lapse movie showing the appearance of CFP-PCNA foci during S-phase.

---

replication of the region is completed. However, in the majority of cells, the time between fork blockage and convergence will range from 10 to 60 min, which likely accounts for the variable duration of Rad52 focus co-localization with *lacO*-LacI in cells with *RTS1-AO*.

To determine what influence the timing of fork convergence has on both the initiation and progression of HR at a blocked replication fork, we deleted *ori-1253*, which is the most efficient origin

**Table 2**. Distances and estimated replication times from origin to *RTS1*

|  | Origin | Distance from barrier (bp) | Time from barrier (min) |
|---|---|---|---|
| Origin centromere proximal relative to *RTS1* | ori-1141 | 186,731 | 62.2 |
|  | ori-1194 | 134,208 | 44.7 |
|  | ori-1253 | 75,154 | 25.1 |
|  | ori-1277 | 51,031 | 17.0 |
| Origin centromere distal relative to *RTS1* | ori-1323 | 7735 | 2.6 |
|  | ori-1338 | 23,098 | 7.7 |
|  | ori-1342 | 27,172 | 9.1 |
|  | ori-1371 | 55,444 | 18.5 |

Distances are calculated from the midpoint of the origin coordinates as stated in OriDB (pombe.oridb.org). Times are based on a replication fork velocity of 3.0 kb/min.

on the centromere-proximal side of *RTS1* firing in 84% of S-phases (*Daigaku et al., 2015*). If forks emanating from this origin prevent and curtail HR at *RTS1-AO*, then there should be an increase in both the frequency and duration of Rad52 foci co-localizing with *lacO*-LacI. Indeed, this is exactly what we observed (*Figure 6B,C*, *Figure 6—figure supplement 1*). The percentage of *RTS1-AO*-containing cells exhibiting a Rad52 focus co-localizing with *lacO*-LacI increased from ~60 to ~80%, and the average duration of these foci increased from 3.4 time points (12 to <17 min) to 5.8 (24 to <29 min) (*Figure 6B*, *Figure 6—figure supplement 2*). However, the timing of Rad52 recruitment to *RTS1-AO* was unaltered indicating that early fork convergence does not mask the detection of recombination that initiates faster than documented in *Figure 5B,C* (*Figure 6C*).

To determine whether the increased frequency and duration of Rad52 co-localization with *lacO*-LacI correlates with an increase in HR, we compared the frequency of direct-repeat recombination in wild-type and *ori-1253Δ* strains containing either *RTS1-IO* or *RTS1-AO* (*Figure 6D*, *Table 1*). In *ori-1253Δ* cells containing *RTS1-IO*, there is a modest ~1.5-fold increase in the very low frequency of spontaneous *ade+* recombinants, whereas in cells with *RTS1-AO* the already high frequency of recombinants increases by ~2.3-fold. Together, these data indicate that recombination operates in the interval between fork blockage and convergence, and if this is too short there will be insufficient time for recombination proteins to act. Moreover, we can conclude that fork convergence is capable of terminating ongoing recombination and does not itself normally induce HR.

## Restarted replication forks suffer frequent collapse and further rounds of recombination

Replication forks restarted following blockage at *RTS1* are prone to perform a U-turn at small inverted repeats positioned within 2.4 kb downstream of the barrier (*Mizuno et al., 2013*). To determine whether this property of the restarted fork is due to a tendency for it to collapse and undergo further rounds of recombination, we positioned our *ade6−* direct repeat recombination reporter 0.2 kb downstream of *RTS1* (*Figure 7A*; site A) and measured the frequency of *ade+* recombinants (*Figure 7B*). With *RTS1-IO* the recombinant frequency was similar to the background level of spontaneous recombination, whereas with *RTS1-AO*, it increased by ~134-fold and ~3.3-fold more than when the barrier is positioned between the *ade6−* repeats (*Figure 7B*, *Table 1*). More than 80% of these recombinants are deletions; however, gene conversions also increase substantially by ~89-fold over spontaneous levels. This is significant because, unlike deletions that can be formed without Rad51, gene conversions depend on Rad51 catalysing the invasion of a donor duplex by a homologous ssDNA tail (*Doe et al., 2004*; *Lorenz et al., 2009*). Therefore, our data indicate that the restarted fork must suffer disengagement of the elongating nascent strands to facilitate HR. This is reminiscent of BIR, which is prone to template switching within the first 10 kb from the DSB (*Smith et al., 2007*).

To see whether forks restarted from *RTS1* remain prone to recombination as they progress further from the RFB, we inserted the *ade6−* direct repeat 12.4 kb downstream of *RTS1*

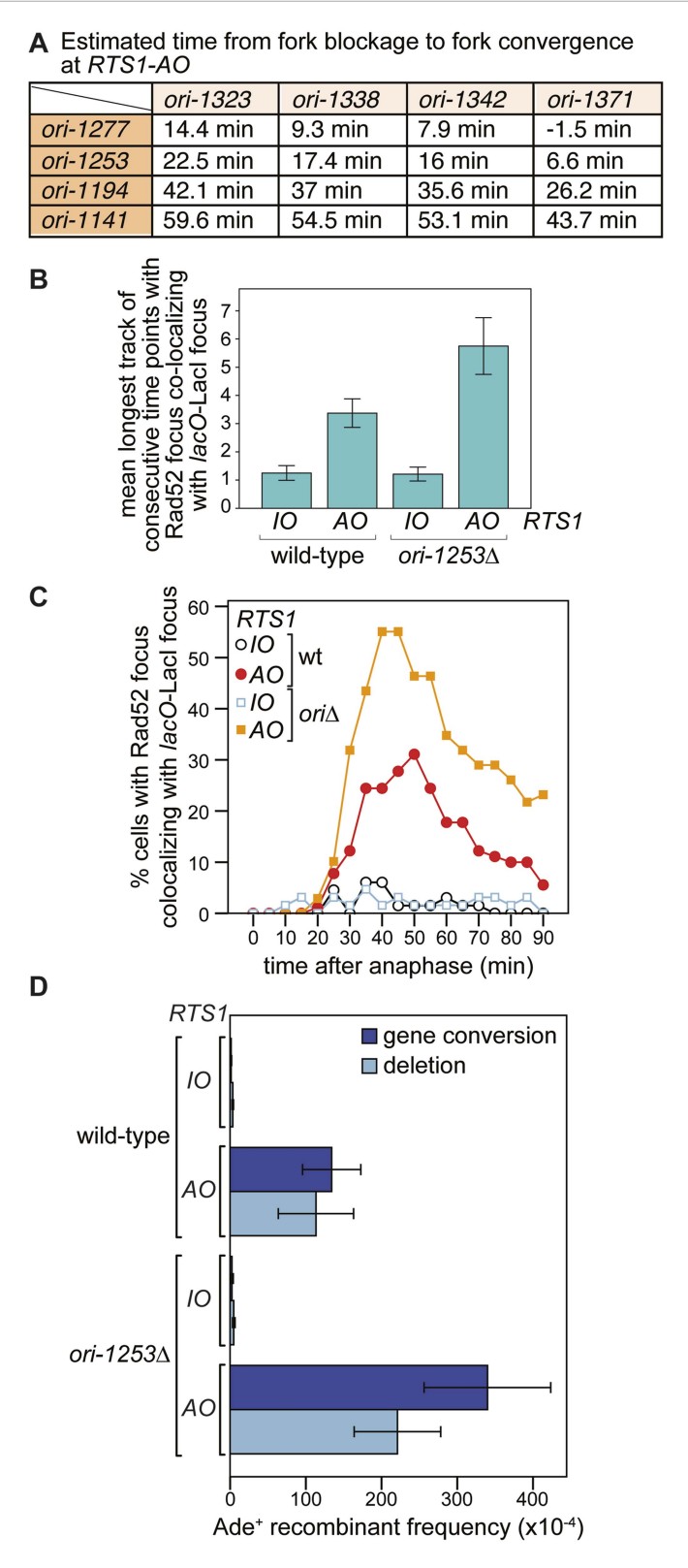

**A** Estimated time from fork blockage to fork convergence at *RTS1-AO*

|  | ori-1323 | ori-1338 | ori-1342 | ori-1371 |
|---|---|---|---|---|
| ori-1277 | 14.4 min | 9.3 min | 7.9 min | -1.5 min |
| ori-1253 | 22.5 min | 17.4 min | 16 min | 6.6 min |
| ori-1194 | 42.1 min | 37 min | 35.6 min | 26.2 min |
| ori-1141 | 59.6 min | 54.5 min | 53.1 min | 43.7 min |

**Figure 6**. The effect of deleting *ori-1253* on recombination at *RTS1*. (**A**) Estimated times from fork blockage at *RTS1-AO* to fork convergence at *RTS1-AO*. Times are calculated from the distance between the midpoint of each origin to *RTS1* (*Siow et al., 2012*) and a fork velocity of 3 kb/min, and it is assumed that each origin fires at the same time

*Figure 6. continued on next page*

*Figure 6. Continued*

during S-phase. (**B**) Effect of deleting *ori-1253* on the duration of Rad52-YFP focus co-localization with the *lacO*-LacI focus. Data are represented as mean ± SEM. (**C**) Effect of deleting *ori-1253* on the percentage of cells with a Rad52-YFP focus co-localizing with the *lacO*-LacI focus during the 90 min post-anaphase. The data in **B** and **C** are derived from *Figure 3C* and *Figure 6—figure supplement 1*. (**D**) *ade*+ recombinant frequencies for strains MCW4712, MCW4713, MCW6894, and MCW6778. Data are represented as mean ± SD.

The following figure supplements are available for figure 6:

**Figure supplement 1**. Tracking Rad52-YFP focus localization at *RTS1* by time-lapse microscopy in cells in which *ori-1253* has been deleted.

**Figure supplement 2**. The duration of Rad52-YFP focus co-localization with the *lacO*-LacI focus in cells with *RTS1-AO* is extended when *ori-1253* is deleted.

---

(*Figure 7A*; site B) and measured the frequency of *ade*+ recombinants (*Figure 7B* and *Table 1*). While the recombinant frequency of *RTS1-IO* was again at background levels, with *RTS1-AO* it increased by 27-fold. Similar to when the direct repeat is at site A, the majority (~94%) of recombinants are deletions, however, gene conversions do increase by ~sixfold over spontaneous levels implying that at least some recombination is Rad51 dependent. Template switching associated with restarted replication has been shown to decrease with distance from the point of initiation (*Smith et al., 2007*; *Mizuno et al., 2013*), and therefore, it seemed likely that the decline in recombination frequency from site A to site B was a consequence of the maturation of the restarted fork as it progressed from the *RTS1* barrier site. However, it was also possible that replication fork convergence was preventing some restarted forks from progressing as far as site B. To investigate this, we measured the recombinant frequency at site B in strains in which *ori-1253* was deleted (*Figure 7B*, *Table 1*). Surprisingly, with *RTS1-AO*, the frequency of gene conversions increased by 54-fold and deletions by 225-fold compared to *RTS1-IO* levels, which also represents an overall ~1.5-fold increase in recombinants compared to site A. These data indicate that restarted forks remain liable to HR over a distance of at least 12.4 kb from their point of initiation with relatively little or no reduction in template switching.

## RDR is not always required for cell viability following replication fork blockage at *RTS1*

Previous studies have shown that both Rad51 and Rad52 are required to maintain cell viability following replication fork blockage at *RTS1* positioned at the *ura4* locus on chromosome 3 (*Lambert et al., 2005*, *2010*). To see if the same is true when *RTS1* is positioned at the *ade6* locus, we compared the viability of wild type and *rad51Δ rad52Δ* double mutant strains harbouring either *RTS1-IO* or *RTS1-AO* in place of the direct repeat recombination reporter at the *ade6* locus (*Figure 8A,B*). Neither wild type nor mutant exhibited a reduction in growth or viability with *RTS1-AO* compared to *RTS1-IO*. We also tested whether delaying fork convergence at *RTS1* by deleting *ori-1253* might necessitate HR, but again no difference was observed in the growth or viability of *RTS1-IO* and *RTS1-AO*-containing strains (*Figure 8A,B*). These data show that the rapid induction of recombination following replication fork blockage at *RTS1* is surprisingly unnecessary for cell viability.

## Discussion

We have shown that HR can be rapidly initiated at a unidirectional RFB seemingly as a default response to restart replication, even when replication by an opposing fork is sufficient to maintain cell viability. There are many examples of weaker RFBs, which cause fork stalling without inducing HR either because the block is quickly removed by an accessory helicase or the intra-S-phase checkpoint is activated to stabilize the replisome and thereby prevent fork collapse (*Azvolinsky et al., 2006*; *Errico and Costanzo, 2012*; *Sabouri et al., 2012*; *Steinacher et al., 2012*). However, in the case of *RTS1*, the accessory helicase Pfh1 fails to dislodge the barrier (*Steinacher et al., 2012*), and the intra-S-phase checkpoint seemingly remains inactive resulting in fork collapse (*Lambert et al., 2005*). Importantly, in wild-type cells breakage of forks blocked at *RTS1* is not a frequent event and therefore

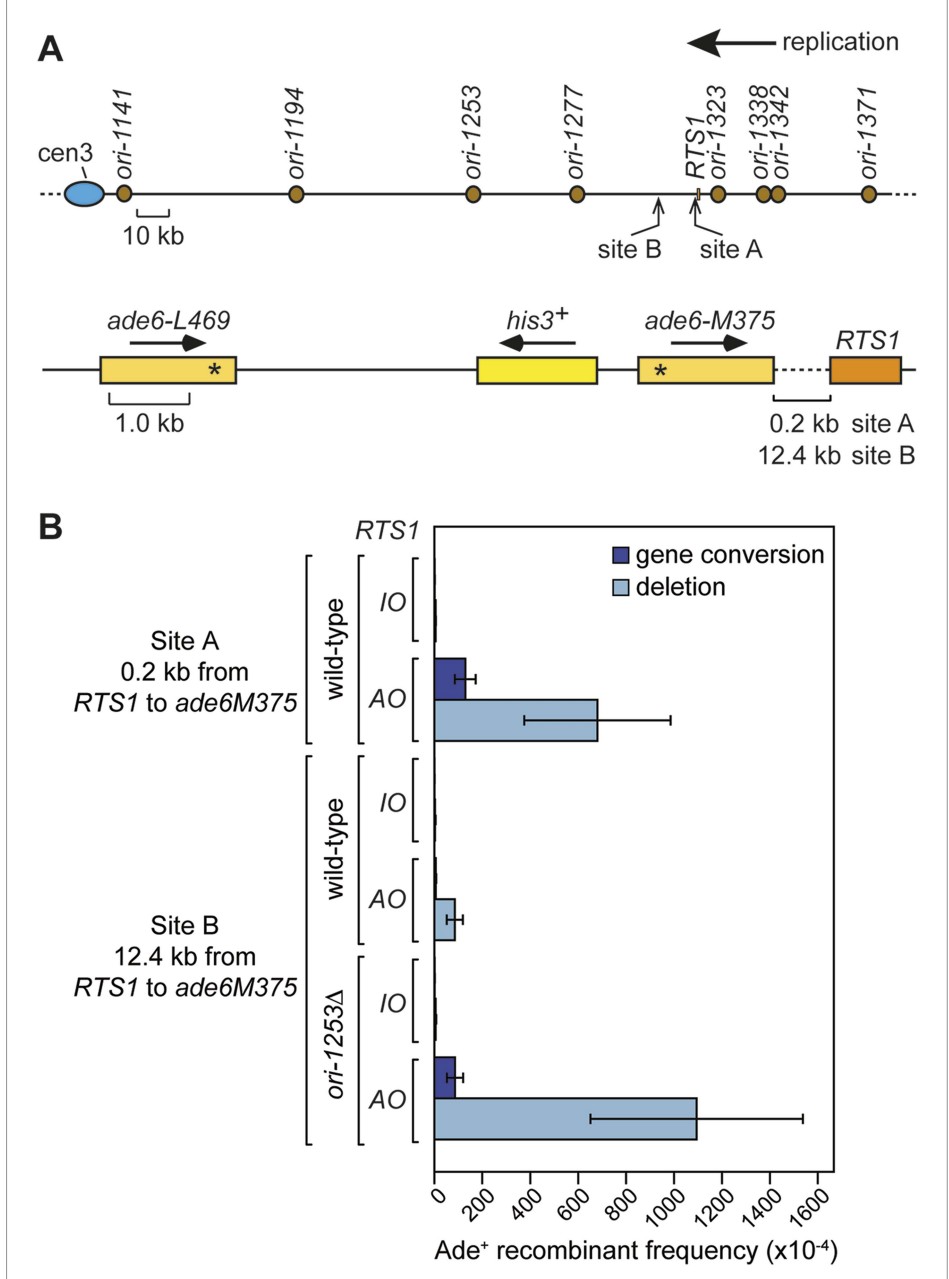

**Figure 7**. Direct repeat recombination downstream of *RTS1*. (**A**) Schematic showing the two positions on chromosome 3 where the *ade6⁻* direct repeat recombination reporter (shown in the bottom panel) is inserted downstream of *RTS1*. (**B**) Ade⁺ recombinant frequencies for strains MCW7131, MCW7133, MCW7257, MCW7259, MCW7293, and MCW7295. Data are represented as mean ± SD.

cannot account for the recruitment of Rad52 in the majority of cells (*Ahn et al., 2005*; *Mizuno et al., 2009*; *Lambert et al., 2010*). Additional evidence that fork breakage is an uncommon event at *RTS1* comes from unpublished data showing that replication fork breakage induced by a site-specific single-strand break, placed at the same position as *RTS1*, causes a quite different frequency of recombinants in a *rad51Δ* mutant than *RTS1-AO*. In the case of *RTS1-AO rad51Δ* causes approximately a 50% reduction in deletions (*Ahn et al., 2005*), whereas with a site-specific single-strand break it causes a 10-fold increase (unpublished data). We suspect that the residual deletions that form in *rad51Δ RTS1-AO* cells stem from Rad52-mediated 'strand invasion' (similar to Rad51-independent BIR in

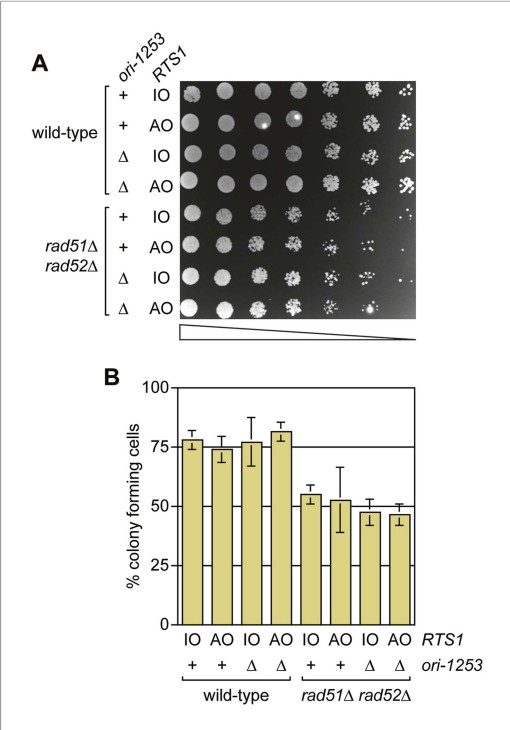

**Figure 8**. A comparison of the growth and viability of wild-type and *rad51Δ rad52Δ* mutant strains containing either *RTS1-IO* or *RTS1-AO* with and without deletion of *ori-1253*. (**A**) Spot assay and (**B**) colony forming assay. The strains are MCW7224, MCW7223, MCW7277, MCW7279, MCW7368, MCW7370, MCW7372, and MCW7374. The data in B are mean values ± SD.

budding yeast [*Anand et al., 2013*]) and/or DSB-independent single-strand annealing that could occur during fork convergence.

## The temporal kinetics of RFB-induced RDR

Detailed analysis of the kinetics of DSB repair by HR at the mating-type locus in *S. cerevisiae* has shown that Rad51 recruitment to DNA begins about 10–20 min after break formation with strand invasion occurring a further 10 min later (*Hicks et al., 2011*). This is broadly in line with the kinetics of Rad52 focus formation following exposure of cells to γ-irradiation, which occurs within 30 min of break formation (*Barlow and Rothstein, 2009*). However, it takes a further 40 min for DNA synthesis to start, which suggests that the assembly of the replication machinery is quite slow (*Hicks et al., 2011*). In the case of BIR, DNA synthesis is delayed even longer (a further 2–4 hr) by a recombination execution checkpoint (*Malkova et al., 2005*; *Jain et al., 2009*). We estimate that Rad52-YFP focus formation can occur as early as 7.4–12.4 min after replication fork blockage at *RTS1*, which is comparable with the kinetics of DSB repair in *S. cerevisiae*, bearing in mind that Rad52 precedes Rad51 in loading onto RPA-coated ssDNA. It is likely that the blocked replication fork would need to undergo remodelling, including DNA strand resection, prior to Rad52 recruitment and therefore RDR probably initiates even earlier.

Although we have not directly measured the timings of the next phases of RDR (i.e., strand invasion and DNA synthesis), we can estimate an upper limit for the total time that these would take based on the heightened direct repeat recombination at sites A and B downstream of *RTS1-AO* being an indicator of restarted fork progression (*Figure 7*). Our observation that deleting *ori-1253* dramatically increases recombinant frequency at site B indicates that progression of the restarted fork is constrained by fork convergence. Therefore, the fact that we detect recombination at sites A and B means that some restarted forks reach these sites before the opposing fork emanating from one of the four centromere proximal origins (assuming that at least one of these origins fires in every cell cycle) (*Figure 7A*). The most distant of these origins (*ori-1141*) lies ~186 kb away from site A and ~174 kb away from site B, which means that RDR would have up to ~60 min from the point of fork blockage at *RTS1* to reach these sites, assuming a normal replication fork velocity of 3 kb/min and similar firing times for the origins that flank the RFB. Based on these estimates, it would appear that the overall temporal kinetics of RFB-induced RDR in *S. pombe* are either similar or faster than those of DSB-induced HR at the mating-type locus in *S. cerevisiae*, and much faster than BIR, presumably due to the absence of a recombination execution checkpoint.

## A hypothetical model for RFB-induced RDR

We have previously proposed that reversal of the collapsed replication fork together with strand resection would generate a ssDNA tail onto which RPA, Rad52, and Rad51 would sequentially load (*Sun et al., 2008*) (*Figure 9A*). Strand invasion would follow, creating a D-loop onto which replication proteins assemble and commence DNA synthesis. As discussed above, we estimate that this whole process would take between ~10 and 60 min. Previous studies have shown that RFB-induced RDR is prone to template slippage/switching (*Iraqui et al., 2012*; *Mizuno et al., 2013*), and based on our work we can conclude that a major part of this infidelity must derive from an instability of the restarted

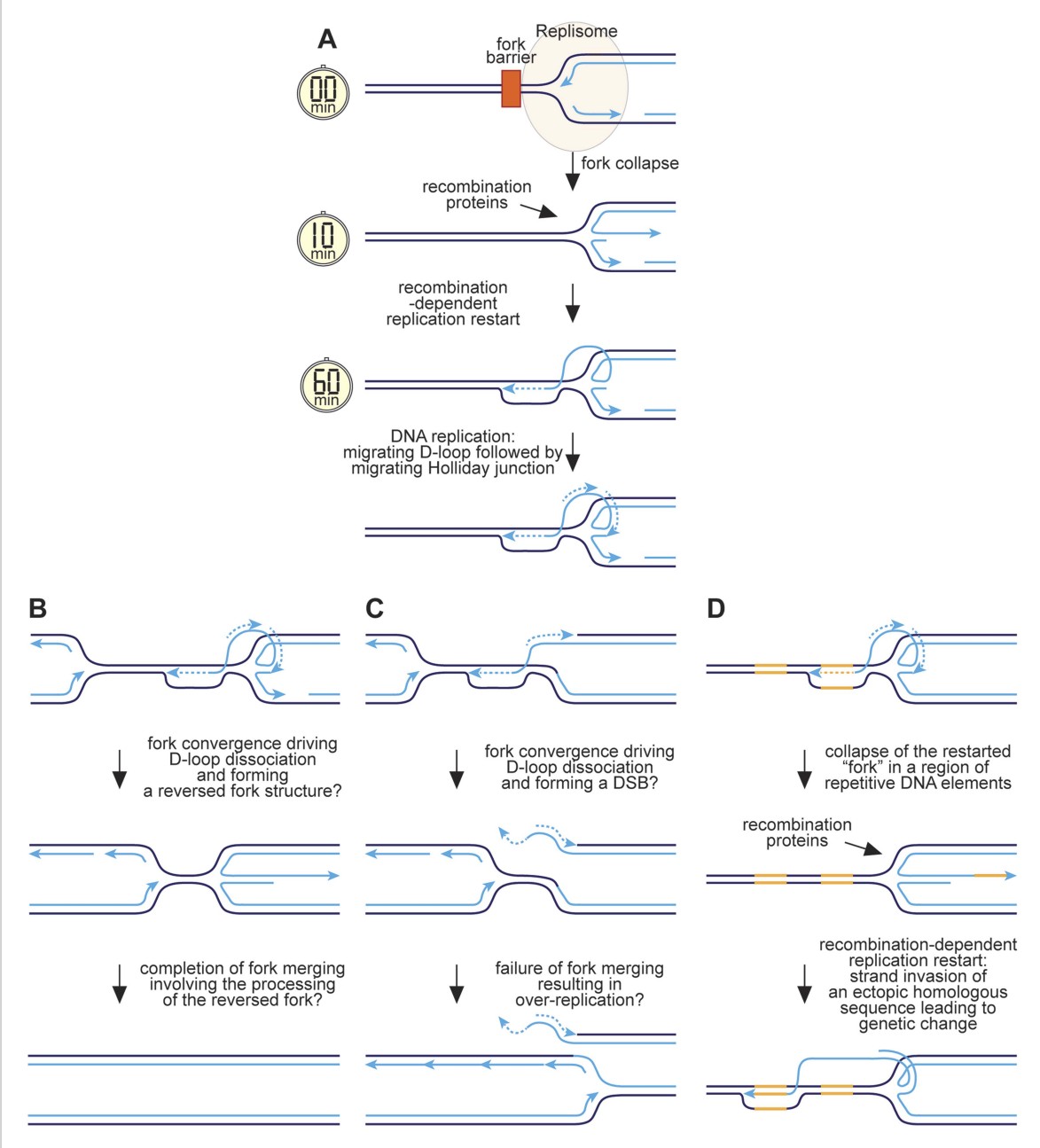

**Figure 9**. Hypothetical model for RFB-induced RDR. (**A**) RDR initiates within 10 minutes of replication fork blockage at *RTS1* and gives rise to a restarted fork within 60 min. Parental DNA strands are shown in dark blue and nascent strands in light blue with light blue arrows indicating the direction of DNA synthesis. (**B**) Model for how a regular and restarted fork might merge. (**C**) Model for how convergence between a regular fork and a restarted fork could lead to over-replication. (**D**) Model for how RDR might give rise to genetic change. The two patches of DNA highlighted in yellow represent a direct repeat of homologous sequences. See main text for further details.

fork, which enables the frequent engagement of HR proteins such as Rad51 that can then catalyse ectopic recombination. We suspect that this is because mechanistically RFB-induced RDR is similar to BIR involving a migrating bubble or D-loop at which conservative replication occurs (*Saini et al., 2013*; *Wilson et al., 2013*) (*Figure 9A*). One key difference, however, is that the invading DNA strand may remain connected to the reversed fork structure from where it originated. Lagging strand synthesis would convert the reversed fork into a fully ligated four-way DNA/Holliday junction, which could branch migrate behind the D-loop. This might be advantageous during convergence with a normal

replication fork, which could drive dissociation of the D-loop through the action of the replisome's accessory DNA helicase Pfh1 translocating on the lagging template strand, thereby converting the restarted 'fork' back into a reversed fork at which fork merging could occur (*Figure 9B*). In contrast, early resolution of the Holliday junction could result in a one-ended DSB during fork convergence and even a failure of fork termination leading to over-replication (*Figure 9C*).

The presence of a Holliday junction migrating behind the D-loop might also account for the tendency of the latter to 'collapse' if there is a failure to coordinate migration rates. For example, any impedance of the D-loop's migration might result in the Holliday junction 'catching up' with it, driving its dissociation and thereby reforming a reversed fork. This would allow the reloading of HR proteins, which could then catalyse ectopic recombination at sites distant from the original RFB (*Figure 9D*). The high frequency of ectopic recombination downstream of the RFB (*Figure 5B*) suggests that dissociation of the putative migrating D-loop within *ade6-M375* frequently gives rise to strand invasion of *ade6-L469*. The high proportion of deletions suggests that the newly established D-loop tends to migrate beyond *ade6-L469* rather than dissociating a second time to trigger the possibility of re-invasion into *ade6-M375* and the formation of a *ade6+* gene conversion. The extended tracks of Rad52 focus co-localization with *lacO*-LacI, which are especially prevalent in *ori-1253∆* cells and can last for more than 60 min (*Figure 6—figure supplement 2*), might be representative of multiple rounds of recombination caused by the frequent collapse of the restarted 'fork' as it moves away from *RTS1*. However, it is also possible that Rad52 remains associated with the restarted 'fork', enabling more rapid recovery of any subsequent collapse.

## RDR as a default response to replication fork collapse

From an evolutionary standpoint, the rapid induction of HR following replication fork collapse may make sense if no checkpoint response is induced, as it would ensure that RDR commenced without unnecessary delay thereby maximizing the chance that genome duplication is completed in a timely manner. This would be especially important given that the initiation of RDR appears to have some kinetically slow steps. A failure to complete DNA replication prior to mitosis would result in DNA breakage and chromosome missegregation, and may account for pathologies such as micronuclei formation and chromothripsis that can drive cancer development in human cells (*Crasta et al., 2012*; *Beuzer et al., 2014*). Presumably in our experimental system, RDR is unnecessary for maintaining cell viability because passive replication from one of the centromere proximal origins is sufficient to complete replication on time. However, the same may not be true at the *ura4* locus in *S. pombe* where replication fork blockage by *RTS1* imposes a definite requirement for HR to maintain cell viability (*Lambert et al., 2005*, *2010*). Although it should be noted that, unlike in our study, Lambert et al. made use of an inducible promoter to drive the overexpression of the fork barrier protein Rtf1, which might alter the nature of the RFB and the requirement for recombination proteins. One example where RDR most likely is needed to ensure that DNA replication is completed on time is at common fragile site loci in human cells, which replicate late in S-phase and with greater distances between active origins (*Ozeri-Galai et al., 2011*; *Mankouri et al., 2013*). This creates a scenario where the irreversible stalling of two converging replication forks would result in mitosis proceeding with a region of unreplicated DNA (*Mankouri et al., 2013*). As discussed by others, the consequences of this may be far worse than the risks associated with RDR.

## Materials and methods

### Strains and plasmids

*S. pombe* strains and PCR primers are listed in *Tables 3 and 4*, respectively. Plasmids pMN5 and pMN6 were used for the targeted integration of *RTS1-IO* and *RTS1-AO*, respectively, with a nearby 4.5 kb *lacO* array at the *ade6* locus. These plasmids are derivatives of pMW700 and pMW701 (*Ahn et al., 2005*) containing a 4.5 kb blunt-ended XhoI *lacO* array fragment from pLAU43 (*Lau et al., 2003*) inserted at a PvuII site 5′ of *his3*. Plasmid pMN7 is a derivative of pAG32 (*Goldstein and McCusker, 1999*) and was used for the targeted integration of *nmt41-NLS-lacI-tdKatushka2-hphMX4* at the *lys1* locus. Plasmids pMW921, pMW922, pMJ20, and pMJ21 are also derivatives of pAG32 with *RTS1* from pRS115 (*Steinacher et al., 2012*) inserted at its BamHI site. pMW921 and pMW922 contain DNA fragments from the genomic regions flanking *ade6* to facilitate its targeted deletion and

replacement with *RTS1-AO-hphMX4* and *RTS1-IO-hphMX4*, respectively. The fragments were amplified using primers oMW1625 + oMW1626 and oMW1627 + oMW1628 and cloned into the HindIII-SalI and SacI-SpeI sites in pAG32, respectively. pMJ20 and pMJ21 contain DNA fragments from the genomic region 3′ of *ade6* to facilitate targeted integration of *RTS1-IO-hphMX4* (pMJ20) and *RTS1-IO-hphMX4* (pMJ21) ~0.2 kb 3′ of the stop codon in *ade6*. The fragments were amplified using primers oMW1563 + oMW1564 and oMW1565 + oMW1566 and cloned into the HindIII-SalI and SacI-SpeI sites in pAG32, respectively. pBU2 is a derivative of pAG25 (*Goldstein and McCusker, 1999*) used for the targeted deletion of *ori-1253*. It contains DNA fragments from the genomic regions flanking *ori-1253* amplified using primers oMW1577 + oMW1578 and oMW1579 + oMW1580 and cloned into the HindIII-BamHI and SacI-EcoRI sites in pAG25, respectively. To integrate the *ade6⁻* direct repeat recombination reporter ~12.4 kb away from the normal *ade6* locus, we first constructed a derivative of pFA6-KanMX6 (*Bähler et al., 1998*) (pMW923) containing *ade6-M375-kanMX6* flanked by DNA fragments amplified from genomic DNA using primers oMW1617 + oMW1618 and oMW1619 + oMW1620. Following integration of *ade6-M375::kanMX6* in a *ade6Δ* strain, insertion of the *ade6⁻* direct repeat recombination reporter was achieved by transfomation with BlpI-linearized pFOX2 as described previously (*Osman et al., 2000*). Targeted replacement of *rad51⁺* with *ECFP-rad51⁺-kanMX6* was achieved using a derivative of pFA6a-KanMX6, pMW624. A cassette containing *ECFP-rad51⁺-kanMX6* was amplified from this plasmid using primers oMW627 + oMW628 to incorporate *rad51* 5′ and 3′ UTR sequences to facilitate gene targeting by homologous recombination. The *kanMX6* was later replaced with *arg3⁺* by marker swapping. Strains in which *rad51⁺* is replaced with *ECFP-rad51⁺-kanMX6* exhibit genotoxin sensitivity at a level that is intermediate between wild-type and a *rad51Δ* mutant (unpublished data). To overcome this, we inserted a copy of *rad51⁺* at the *ura4* locus by gene targeting using a EcoRV-DraIII fragment from pMW875. pMW875 was made by amplifying *rad51⁺*, together with its 5′ and 3′ UTRs, from genomic DNA using primers oMW1257 + oMW1258 and cloning this into a StuI site in *ura4⁺* in pREP42 (*Basi et al., 1993*). Plasmids were verified by DNA sequencing and strains were verified by diagnostic PCR, DNA sequencing, and Southern blot analysis where necessary.

## Media and genetic methods

Standard protocols were used for the growth and genetic manipulation of *S. pombe* (*Moreno et al., 1991*). The complete and minimal media were yeast extract with supplements (YES) and Edinburgh minimal medium plus 3.7 mg/ml sodium glutamate (EMMG) and appropriate amino acids (0.2475 mg/ml), respectively. Ade⁺ recombinants were selected on YES lacking adenine and supplemented with 200 mg/l guanine to prevent uptake of residual adenine. Indicated amounts of thiamine were included where appropriate to repress expression from the *nmt* promoter.

## Microscopy

Live cells were mounted with soybean lectin (Sigma-Aldrich, St Louis, MO) in a glass bottom culture dish (MatTek Corp., Ashland, MA) containing liquid media (EMMG with 1 $\mu$M thiamine and without histidine) in a thermally insulated temperature-controlled chamber at 30˚C. Cells were imaged on an inverted Olympus IX71 microscope (Tokyo, Japan) controlled by a DV Elite Core using DeltaVision softWoRx 5.5.0 software using CFP/YFP/mCherry filters (Applied Precision Inc., Issaquah, WA). Images were taken with an Evolve 512 EMCCD camera (Photometrics, Tucson, AZ) using an oil-immersed Olympus 100X UPlanSApo objective with a NA of 1.40. A stack of 16 focal planes at a step-size of 0.3 μm was taken for 'snapshot' analysis. For time-lapse analysis, stacks were taken every 5 min for up to 4 hr. Brightfield images of cells and LacI-tdKatushka2 were imaged with a 100 ms exposure time and 10% neutral density filter. Rad52-YFP was imaged with a 100 ms exposure time and either 5% (time-lapse) or 10% (snapshot) neutral density filter, CFP-PCNA with a 25 ms exposure time and 2% neutral density filter, ECFP-Rad51 with a 200 ms exposure time and 10% neutral density filter, and Rad54-GFP with a 100 ms exposure time and 10% neutral density filter.

## Image processing and analysis

Images from the DV Elite Core were denoised using a patch-based denoising algorithm (*Kervrann and Boulanger, 2006*) and deconvolved with softWoRx 5.5.0 software (Applied Precision, Issaquah, WA). Foci were scored manually and separately for each fluorescent channel. Foci were distinguished as being a minimum of three to five times brighter than background intensity levels with a minimum of a 2×2 pixel volume. Co-localization was manually assessed and scored as foci in separate channels

**Table 3**. List of *S. pombe* strains used in this study

| Strain | Mating type | Genotype | Source |
|---|---|---|---|
| MCW429 | *h+* | *ade6-M375 int::pUC8/his3+/ade6-L469 ura4-D18 his3-D1 leu1-32 arg3-D4* | Lab strain |
| MCW1687 | *h+* | *rad52Δ::ura4+ ade6-M375 int::pUC8/ his3+/RTS1-IO/ade6-L469 ura4-D18 his3-D1 leu1-32 arg3-D4* | Lab strain |
| MCW1688 | *h+* | *rad52Δ::ura4+ ade6-M375 int::pUC8/ his3+/RTS1-AO/ade6-L469 ura4-D18 his3-D1 leu1-32 arg3-D4* | Lab strain |
| MCW1696 | *h+* | *rad51Δ::arg3+ rad52Δ::ura4+ ade6-M375 int::pUC8/his3+/RTS1-AO/ade6-L469 ura4-D18 his3-D1 leu1-32 arg3-D4* | Lab strain |
| MCW4712 | *h+* | *ade6-M375 int::pUC8/his3+/RTS1-IO/ ade6-L469 ura4-D18 his3-D1 leu1-32 arg3-D4* | Lab strain |
| MCW4713 | *h+* | *ade6-M375 int::pUC8/his3+/RTS1-AO/ ade6-L469 ura4-D18 his3-D1 leu1-32 arg3-D4* | Lab strain |
| MCW6298 | *h+* | *ade6-M375 int::pUC8/lacO115/his3+/ RTS1-IO/ade6-L469 ura4-D18 his3-D1 leu1-32 arg3-D4* | This study |
| MCW6302 | *h+* | *ade6-M375 int::pUC8/lacO115/his3+/ RTS1-AO/ade6-L469 ura4-D18 his3-D1 leu1-32 arg3-D4* | This study |
| MCW6351 | *h+* | *ade6-M375 int::pUC8/lacO115/his3+/ RTS1-IO/ade6-L469 lys1-::Pnmt41-NLS-lacI-tdKatushka2-hphMX4 ura4-D18 his3-D1 leu1-32 arg3-D4* | This study |
| MCW6395 | *h+* | *ade6-M375 int::pUC8/lacO115/his3+/ RTS1-IO/ade6-L469 lys1-::Pnmt41-NLS-lacI-tdKatushka2-hphMX4 rad52+::YFP-kanMX6 ura4-D18 his3-D1 leu1-32 arg3-D4* | This study* |
| MCW6536 | *h+* | *ade6-M375 int::pUC8/lacO115/his3+/ RTS1-AO/ade6-L469 lys1-::Pnmt41-NLS-lacI-tdKatushka2-hphMX4 ura4-D18 his3-D1 leu1-32 arg3-D4* | This study |
| MCW6556 | *h+* | *ade6-M375 int::pUC8/lacO115/his3+/ RTS1-AO/ade6-L469 lys1-::Pnmt41-NLS-lacI-tdKatushka2-hphMX4 rad52+::YFP-kanMX6 ura4-D18 his3-D1 leu1-32 arg3-D4* | This study* |
| MCW6701 | *h+* | *ade6-M375 int::pUC8/lacO115/his3+/ RTS1-IO/ade6-L469 lys1-::Pnmt41-NLS-lacI-tdKatushka2-hphMX4 ura4-::pECFP-PCNA+ his3-D1 leu1-32 arg3-D4* | This study† |
| MCW6706 | *h+* | *ade6-M375 int::pUC8/lacO115/his3+/ RTS1-AO/ade6-L469 lys1-::Pnmt41-NLS-lacI-tdKatushka2-hphMX4 ura4-::pECFP-PCNA+ his3-D1 leu1-32 arg3-D4* | This study† |
| MCW6712 | *h+* | *ade6-M375 int::pUC8/lacO115/his3+/ RTS1-IO/ade6-L469 lys1-::Pnmt41-NLS-lacI-tdKatushka2-hphMX4 rad52+::YFP-kanMX6 ura4-::pECFP-PCNA+ his3-D1 leu1-32 arg3-D4* | This study*·† |
| MCW6778 | *h+* | *oriIII-1253Δ::natMX4 ade6-M375 int:: pUC8/his3+/RTS1-AO/ade6-L469 ura4-D18 his3-D1 leu1-32 arg3-D4* | This study |
| MCW6780 | *h+* | *oriIII-1253Δ::natMX4 ade6-M375 int:: pUC8/lacO115/his3+/RTS1-AO/ade6-L469 lys1-::Pnmt41-NLS-lacI-tdKatushka2-* | This study* |

*Table 3. Continued on next page*

Table 3. Continued

| Strain | Mating type | Genotype | Source |
|---|---|---|---|
| | | hphMX4 rad52⁺::YFP-kanMX6 ura4-D18 his3-D1 leu1-32 arg3-D4 | |
| MCW6894 | h⁺ | oriIII-1253Δ::natMX4 ade6-M375 int:: pUC8/his3⁺/RTS1-IO/ade6-L469 ura4-D18 his3-D1 leu1-32 arg3-D4 | This study |
| MCW6902 | h⁻ | oriIII-1253Δ::natMX4 ade6-M375 int:: pUC8/lacO¹¹⁵/his3⁺/RTS1-IO/ade6-L469 lys1⁻::Pnmt41-NLS-lacI-tdKatushka2-hphMX4 rad52⁺::YFP-kanMX6 ura4-D18 his3-D1 leu1-32 arg3-D4 | This study* |
| MCW7065 | h⁺ | ade6-M375 int::pUC8/lacO¹¹⁵/his3⁺/ RTS1-AO/ade6-L469 lys1⁻::Pnmt41-NLS-lacI-tdKatushka2-hphMX4 rad52⁺::YFP-kanMX6 ura4::pECFP-PCNA⁺ his3-D1 leu1-32 arg3-D4 | This study*,† |
| MCW7111 | h⁺ | ade6-M375 int::pUC8/his3⁺/RTS1-IO/ ade6-L469 rad52⁺::YFP-kanMX6 ura4-D18 his3-D1 leu1-32 arg3-D4 | This study* |
| MCW7114 | h⁺ | ade6-M375 int::pUC8/his3⁺/RTS1-AO/ ade6-L469 rad52⁺::YFP-kanMX6 ura4-D18 his3-D1 leu1-32 arg3-D4 | This study* |
| MCW7131 | h⁺ | ade6-M375 int::pUC8/his3⁺/ade6-L469/ RTS1-IO/ hphMX4 ura4-D18 his3-D1 leu1-32 arg3-D4 | This study |
| MCW7133 | h⁺ | ade6-M375 int::pUC8/his3⁺/ade6-L469/ RTS1-AO/ hphMX4 ura4-D18 his3-D1 leu1-32 arg3-D4 | This study |
| MCW7257 | h⁺ | ade6Δ::RTS1-IO-hphMX4 (12.4 kb from ade6)int::ade6-M375/pUC8/his3⁺/ade6-L469/kanMX6 ura4-D18 his3-D1 leu1-32 arg3-D4 | This study |
| MCW7259 | h⁺ | ade6Δ::RTS1-AO-hphMX4 (12.4 kb from ade6)int::ade6-M375/pUC8/his3⁺/ade6-L469/kanMX6 ura4-D18 his3-D1 leu1-32 arg3-D4 | This study |
| MCW7293 | h⁺ | oriIII-1253Δ::natMX4 ade6Δ::RTS1-IO-hphMX4 (12.4 kb from ade6)int::ade6-M375/pUC8/his3⁺/ade6-L469/kanMX6 ura4-D18 his3-D1 leu1-32 arg3-D4 | This study |
| MCW7295 | h⁺ | oriIII-1253Δ::natMX4 ade6Δ::RTS1-AO-hphMX4 (12.4 kb from ade6)int::ade6-M375/pUC8/his3⁺/ade6-L469/kanMX6 ura4-D18 his3-D1 leu1-32 arg3-D4 | This study |
| MCW7223 | h⁻ | ade6Δ::RTS1-AO-hphMX4 ura4-D18 his3-D1 leu1-32 arg3-D4 | This study |
| MCW7224 | h⁻ | ade6Δ::RTS1-IO-hphMX4 ura4-D18 his3-D1 leu1-32 arg3-D4 | This study |
| MCW7277 | h⁻ | oriIII-1253Δ::natMX4 ade6Δ::RTS1-IO-hphMX4 ura4-D18 his3-D1 leu1-32 arg3-D4 | This study |
| MCW7279 | h⁻ | oriIII-1253Δ::natMX4 ade6Δ::RTS1-AO-hphMX4 ura4-D18 his3-D1 leu1-32 arg3-D4 | This study |
| MCW7368 | h⁻ ˢᵐᵗ⁰ | rad51Δ::arg3⁺ rad52Δ::ura4⁺ ade6Δ:: RTS1-AO-hphMX4 ura4-D18 his3-D1 leu1-32 arg3-D4 | This study |
| MCW7370 | h⁻ ˢᵐᵗ⁰ | rad51Δ::arg3⁺ rad52Δ::ura4⁺ ade6Δ:: RTS1-IO-hphMX4 ura4-D18 his3-D1 leu1-32 arg3-D4 | This study |

Table 3. Continued on next page

*Table 3. Continued*

| Strain | Mating type | Genotype | Source |
|--------|-------------|----------|--------|
| MCW7372 | $h^-$ smt0 | rad51Δ::arg3$^+$ rad52Δ::ura4$^+$ oriIII-1253Δ::natMX4 ade6Δ::RTS1-IO-hphMX4 ura4-D18 his3-D1 leu1-32 arg3-D4 | This study |
| MCW7374 | $h^-$ smt0 | rad51Δ::arg3$^+$ rad52Δ::ura4$^+$ oriIII-1253Δ::natMX4 ade6Δ::RTS1-AO-hphMX4 ura4-D18 his3-D1 leu1-32 arg3-D4 | This study |
| MCW7638 | $h^+$ | ade6-M375 int::pUC8/lacO$^{115}$/his3$^+$/RTS1-AO/ade6-L469 lys1$^-$::Pnmt41-NLS-lacI-tdKatushka2-hphMX4 rad52$^+$::YFP-kanMX6 rad51$^+$::ECFP-rad51$^+$-arg3$^+$ ura4$^-$::rad51$^+$ his3-D1 leu1-32 arg3-D4 | This study* |
| MCW7640 | $h^+$ | ade6-M375 int::pUC8/lacO$^{115}$/his3$^+$/RTS1-IO/ade6-L469 lys1$^-$::Pnmt41-NLS-lacI-tdKatushka2-hphMX4 rad52$^+$::YFP-kanMX6 rad51$^+$::ECFP-rad51$^+$-arg3$^+$ ura4$^-$::rad51$^+$ his3-D1 leu1-32 arg3-D4 | This study* |
| MCW7645 | $h^+$ | ade6-M375 int::pUC8/lacO$^{115}$/his3$^+$/RTS1-AO/ade6-L469 lys1$^-$::Pnmt41-NLS-lacI-tdKatushka2-hphMX4 rad54$^+$::GFP-kanMX6 ura4-D18 his3-D1 leu1-32 arg3-D4 | This study‡ |
| MCW7646 | $h^+$ | ade6-M375 int::pUC8/lacO$^{115}$/his3$^+$/RTS1-IO/ade6-L469 lys1$^-$::Pnmt41-NLS-lacI-tdKatushka2-hphMX4 rad54$^+$::GFP-kanMX6 ura4-D18 his3-D1 leu1-32 arg3-D4 | This study‡ |

*rad52$^+$::YFP-KanMX6 was derived from SP220 (**Meister et al., 2003**).
†ura4$^-$::pECFP-PCNA$^+$ was derived from SP154 (**Meister et al., 2005**).
‡rad54$^+$::GFP-kanMX6 was derived from TNF3945 (**Maki et al., 2011**).

overlapping in xyz by a minimum of 2 pixels. Fluorescent signals were quantified using Volocity (Improvision, Coventry, England) or Imaris (Bitplane, South Windsor, CT).

## Recombination assays

Direct repeat recombination was assayed by measuring the frequency of Ade$^+$ recombinants as described (*Osman and Whitby, 2009*). Recombinant frequencies represent the mean value from at least 15 colonies for each strain. All statistical analysis was performed on SPSS Statistics (IBM, Armonk, NY). Data were tested for normal distribution by the Shapiro–Wilk normality test. In accordance with the distribution of the data, mean values were compared by the appropriate independent-samples $t$-test. p values below 0.05 were considered significant.

## Colony forming and spot assays

Cells were grown in EMMG at 30°C to mid-exponential phase and were then harvested, washed, and counted using a haemocytometer and resuspended in water at an appropriate density ($1 \times 10^7$ cells per millilitre for the spot assay and $2.5 \times 10^6$ cells per millilitre for the viability assay). For the spot assay, the suspension was serially diluted in fivefold steps to $1 \times 10^3$ cells per millilitre, and a 10 µl aliquot of each suspension was spotted onto a EMMG plate. The plate was photographed after 4 days at 30°C. For the viability assay, suspensions were recounted to confirm cell densities, and then serial dilutions plated on EMMG in triplicate. Colonies were counted on the appropriate dilution plates after 5–6 days growth at 30°C. The assay was repeated three times to obtain mean values for the percentage of colony forming cells.

## 2D gels

Genomic DNA was prepared from asynchronously growing yeast cultures either by mechanical disruption of cells followed by caesium chloride density gradient centrifugation (*Huberman et al., 1987*) (*Figure 1B*, *Figure 1—figure supplement 1* and *Figure 3—figure supplement 1*) or by enzymatic lysis of cells embedded in agarose (*Lambert et al., 2010*) (*Figure 2*). Replication intermediates were enriched for by fractionation of DNA on BND cellulose unless otherwise stated (*Lambert et al., 2010*).

**Table 4**. List of oligonucleotides used in this study

| Oligonucleotide | Sequence (5′–3′) |
| --- | --- |
| oMW627 | TAATATAAAAAACTCTTTTCAATTCCAGAATAGTGATAA TTTCGTGCTTAACAAGTTATAA TGGTGTATCTCGAGTTGG |
| oMW628 | CAAACTCATCCCATAGAATTTGCAAAATAATAAATAAAA ATGAAACGATACTAAAATAAT ATTCGAGCTCGTTTAAACTG |
| oMW1257 | TTATAGGCCTGACCAGTGCTGTTCTCTTG |
| oMW1258 | TTATAGGCCTGCAAGTGGATCCTTAGGCTTC |
| oMW1563 | TATAAAGCTTGAAATTCTAGATTGTAAAATG |
| oMW1564 | TATAGTCGACTTTCAACTGCTTCACAGCAC |
| oMW1565 | TTAGAGCTCCAATATAATATGCTATAAAGC |
| oMW1566 | TATACTAGTCATCTTTTAATAATTGAAGAC |
| oMW1577 | ATAAAGCTTTAACCATCAGGTTATTCTC |
| oMW1578 | TTTGGATCCGAGGAAATCACAAGCATTTCC |
| oMW1579 | TTTGAGCTCTTGTTCAGAGCTAGGATTCG |
| oMW1580 | ATAGAATTCACGACGAAAACGCGGACATTC |
| oMW1617 | TATCAGCTGTACTTATTTACGTACTGTG |
| oMW1618 | AATGTCGACAAATCAAAACGACTAGCAGTC |
| oMW1619 | TATGAGCTCGAGTAGATAGAATTTTGTGC |
| oMW1620 | AATACTAGTGCGCTGTAACTTACCTAC |
| oMW1625 | TATAAAGCTTGGTGGTGAGGTAAACG |
| oMW1626 | TATAGTCGACTGTAGTGAGTTAGTGCGCAG |
| oMW1627 | TTAGAGCTCCGAATAATGTGCTGCGACG |
| oMW1628 | TTAATAACTAGTCTTAATATTGC |

Replication intermediates were run on 2D gels (*Brewer and Fangman, 1987*) using 0.4% and 1.2% agarose for the first and second dimensions, respectively. Gels were Southern blotted, and the blots were probed with the indicated $^{32}$P-labelled probe and then analysed by phosphorimaging using a Fuji FLA3000 and Image Gauge software (*Lorenz et al., 2009*).

## Acknowledgements

We thank: Claire Bryer, Zsofia Novak, Sharon Ruane, and Brittany Ulloa for help in constructing some of the plasmids/strains used in this study; Giuseppe Baldacci and Sarah Lambert for the CFP-PCNA strain; Takuro Nakagawa for the Rad54-GFP strain; and Alexander Lorenz and Micron Oxford for assistance with the microscopy. This work was supported by grant 090767/Z/09/Z from the Wellcome Trust.

## Additional information

### Funding

| Funder | Grant reference | Author |
| --- | --- | --- |
| Wellcome Trust | 090767/Z/09/Z | Matthew C Whitby |

The funder had no role in study design, data collection and interpretation, or the decision to submit the work for publication.

### Author contributions

MON, Conception and design, Acquisition of data, Analysis and interpretation of data, Drafting or revising the article; MJ, CAM, FO, Acquisition of data, Analysis and interpretation of data, Drafting

or revising the article; MCW, Conception and design, Analysis and interpretation of data, Drafting or revising the article

## Additional files

### Major dataset

The following previously published dataset was used:

| Author(s) | Year | Dataset title | Dataset ID and/or URL | Database, license, and accessibility information |
|---|---|---|---|---|
| Siow CC, Nieduszynska SR, Muller CA, Nieduszynski CA | 2012 | Data from: oriDB | pombe.oridb.org | Available at pombe.oridb.org. |

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
