## [Decision Letter]

Thank you for sending your work entitled “Recombination is a rapid response to replication impedance by *RTS1* leading to restarted forks that are prone to collapse” for consideration at *eLife*. Your article has been favourably evaluated by Fiona Watt (Senior editor), a Reviewing editor, and 3 reviewers.

The Reviewing editor and the reviewers discussed their comments before we reached this decision, and the Reviewing editor has assembled the following comments to help you prepare a revised submission.

Nguyen and colleagues have submitted a very elegant study on replication restart by recombination in *S. pombe*. The manuscript is well written and the data is of high quality. In brief, the authors show by live single cell analysis that the Rad52 recombination mediator is recruited to an *RTS1* blocked fork in almost every S-phase cell. However, many of the conclusions of the study are based on the assumption that Rad52 foci reflect ongoing recombination. While this may be true for repair of a DSB, it may not be the case for a transiently stalled/blocked replication fork. Therefore, it remains crucial to evaluate to what extent the observed Rad52 foci reflect ongoing recombinational restart of replication.

Based on these results, the authors propose that recombination-dependent replication (RDR) is a major pathway of replication restart in their system. Surprisingly, however, they find that RDR is dispensable for the viability of cells experiencing a block in replication. Another significant observation is that the RDR observed by authors is highly unstable, which makes it similar to the BIR mechanism.

Although the observations are interesting and important, the collective major comments that need to be addressed prior to publication are:

1) It is essential that you determine whether the observed Rad52 foci reflect ongoing recombinational restart of replication or they are simply associating with the ssDNA at the replication fork. This issue bears on the recruitment of HR proteins and the involvement of HR at stalled forks as explained by your model (Figure 7). You can address this issue by one or both of the additional experiments:

A) Analysing Rad52 foci in a mutant that cannot complete recombination. If the kinetics and frequency of the majority of Rad52 foci remain unchanged in a recombination deficient mutant, then it would indicate that Rad52 foci do not reflect ongoing recombination but perhaps rather Rad52 associating with RPA bound to ssDNA at the stalled fork. If on the other hand, the duration and frequency of the majority of Rad52 foci dramatically increase in a recombination mutant, it would indicate that the observed Rad52 foci reflect ongoing recombination that is required for the fork to recover/restart. It is not trivial to decide which recombination mutant(s) to analyse, because some mutant might have secondary effects; however, one possibility could be to analyze a rad51∆ mutant. A more elegant approach would be to analyze one of the previously described point mutants of Rad52 that are defective in recombination (Mortensen et al. 2002). Ideally, this analysis should be performed in the oriIII-1253∆ background to reduce rescue from the adjacent replication fork.

B) Alternatively, if the HR intermediate proposed is a common structure at stalled forks, it should be possible to see a Rad52-dependent structure by 2D gel analysis, or at least to see a difference in fork progression in wild-type and *rad52∆* cells. Such an experiment would also address the consequences of RAD52 loss; whether there is cell cycle arrest, or increased fork stalling; and whether Rad52 plays more important role in the absence of origin 1277 (the absence of a timely converging fork).

2) The recruitment of Rad52 to stalled/collapsed forks at the *RTS1* locus is an interesting observation. You used a 115 repeat *lacO* array, inserted next to *RTS1*, coupled with expression of LacI as a marker for stalled forks and followed the kinetics of tagged Rad52 foci formation. The only negative of this experimental set up is that *lacO*-LacI was demonstrated by many groups to block a replication fork itself. Thus on top of the *RTS1*, an additional potentially fork blocking region was inserted to follow the kinetics of HR protein recruitment. It seems that insertion of the *lacO* array has little impact on gene conversion frequencies between ade6 alleles next to *RTS1*; however, replication fork progression was not evaluated by 2D gels. It was somewhat surprising that about 20% of cells with *RTS1-IO* (in inactive orientation) show co-localization of LacI and Rad52. This might suggests that *lacO*-LacI indeed itself is at least a weak fork barrier? You need to clarify whether the *lacO*-LacI interaction does or does not act as RFB. It would seem that the best way to address this question is by 2D gel electrophoresis. Such analysis would also permit measurement of the kinetics of fork stalling at *RTS1-AO*, in relationship to the kinetics of Rad52 foci formation. Therefore, this important issue needs to be resolved by a direct method such as, for example, 2D gel electrophoresis.

3) The high (unchanged) viability of recombination-defective yeast cells following replication collapse should be measured by using more quantitative approaches. The results from spot tests measuring viability of cells undergoing replication collapse are not sufficiently compelling (in Figure 6 and in the subsection headed “RDR is not always required for cell viability following replication fork blockage at *RTS1*”).

4) You must show the “data not shown” referred to in the subsection of the Results entitled “Experimental system”. This is not a request for additional experimentation, but rather to provide the data that you already have. In presenting these unpublished data, you need to explain how you estimated, from Figure 1, the following statement: “It also shows that ∼ 50% of forks remain blocked at *RTS1-AO* long enough for replication to be completed by the opposing fork resulting in fork merging at *RTS1*”.

5) The following are important clarifications that are required in the manuscript; none of these clarifications require additional experimentation.

A) Figure 7. It is unclear why you claim that a replication fork converging with a D-loop would dissociate the D-loop. Why should Mcm2-7 migrating on the leading strand template displace an invading/extended 3' strand that is on lagging strand template?

B) While discussing your results, you propose that Rad52 is recruited to the position of fork collapse, but not breakage. However, it remains unclear how you can distinguish between these two events in your experimental conditions. For example, it appears that frequent formation of deletions that was frequently observed in this and also in your previous paper (see [1]) could be explained by SSA repair, and therefore induced by DNA breakage. Importantly, approximately a half of such deletions were independent of Rad51 (1), which is consistent with SSA. In addition, it remains unclear how you explain the increase of the fraction of deletions (which also could be SSA) among secondary recombination events induced by RDR.

C) You often use “majority”, “minority”, “few”, etc., instead of exact numbers. You need to provide actual values throughout the paper to provide some quantitative context for these relative descriptors. Just a few examples are:

i) In the subsection “Live cell imaging of RDR at *RTS1*”: “The majority of cells exhibited a single Rad52 focus” means 51% or 99%. Please provide an exact number.

ii) In the subsection “Timing of Rad52 recruitment following replication fork blockage”: “In a few cells…”. Please provide the number instead of “few”. By eye it looks like at least 1/4 of all cells with co-localization.

iii) The words “rapid” and “early” in the Title and Abstract are not informative, unless it is clear which quantity they being compared to. I suggest rephrasing.

---

## [Author Response]

*1) It is essential that you determine whether the observed Rad52 foci reflect ongoing recombinational restart of replication or they are simply associating with the ssDNA at the replication fork. This issue bears on the recruitment of HR proteins and the involvement of HR at stalled forks as explained by your model (*Figure 7*). You can address this issue by one or both of the additional experiments*:

*A) Analyzing Rad52 foci in a mutant that cannot complete recombination. If the kinetics and frequency of the majority of Rad52 foci remain unchanged in a recombination deficient mutant, then it would indicate that Rad52 foci do not reflect ongoing recombination but perhaps rather Rad52 associating with RPA bound to ssDNA at the stalled fork. If on the other hand, the duration and frequency of the majority of Rad52 foci dramatically increase in a recombination mutant, it would indicate that the observed Rad52 foci reflect ongoing recombination that is required for the fork to recover/restart. It is not trivial to decide which recombination mutant(s) to analyze, because some mutant might have secondary effects; however, one possibility could be to analyze a* rad51∆ *mutant. A more elegant approach would be to analyze one of the previously described point mutants of Rad52 that are defective in recombination (Mortensen et al. 2002). Ideally, this analysis should be performed in the oriIII-1253∆ background to reduce rescue from the adjacent replication fork*.

*B) Alternatively, if the HR intermediate proposed is a common structure at stalled forks, it should be possible to see a Rad52-dependent structure by 2D gel analysis, or at least to see a difference in fork progression in wild type and* rad52∆ *cells. Such an experiment would also address the consequences of RAD52 loss; whether there is cell cycle arrest, or increased fork stalling; and whether Rad52 plays more important role in the absence of origin 1277 (the absence of a timely converging fork)*.

The reviewers make a very valid point. To address this we initially analyzed Rad52 foci in a *rad51∆* mutant as suggested by the reviewers. Contrary to expectation we observed no significant change in the timing or frequency of Rad52 co-localization with *lacO*-LacI when *rad51* was deleted. However, a caveat to this experiment is that the overall frequency and number of non-co-localizing Rad52 foci increases markedly in a *rad51∆* mutant, which could result in less protein being available for recruitment to the *RTS1-AO* site. Consequently we have not included these data in our revised manuscript. Instead we have focused on: 1) analyzing replication intermediates surrounding *RTS1-AO* in a *rad51∆ rad52∆* double mutant by 2D gel analysis (in line with the second experiment proposed by the reviewers); and 2) further imaging experiments to determine whether other recombination proteins in addition to Rad52 are recruited to the barrier. The new 2D gel analysis is presented in Figure 2 and described in a new Results section entitled “Rad52 is required for replication past *RTS1-AO*”. Consistent with Rad51 and/or Rad52 being required for RDR at *RTS1* we observe a loss of large Y-shaped DNA molecules past the barrier in a *rad51∆ rad52∆* double mutant. In the new imaging experiments (presented in Figure 4 and described in a new Results section entitled “Both Rad51 and Rad54 are recruited to *RTS1-AO*” we observe that both Rad51 and Rad54 co-localize with the *lacO*-LacI array adjacent to *RTS1-AO* at similar frequencies as Rad52. Importantly more than 90% of Rad52 foci that co-localize with *lacO*-LacI in *RTS1-AO* cells also co-localize with a Rad51 focus (indicating that Rad52 remains associated with the site after Rad51 has loaded, which could explain why we don’t see an increase in Rad52 foci at the barrier in a *rad51∆* mutant). The presence of three core recombination proteins (Rad51, Rad52 and Rad54) at the *RTS1* barrier provides strong evidence of ongoing recombination activity.

*2) The recruitment of Rad52 to stalled/collapsed forks at the* RTS1 *locus is an interesting observation. You used a 115 repeat* lacO *array, inserted next to* RTS1*, coupled with expression of LacI as a marker for stalled forks and followed the kinetics of tagged Rad52 foci formation. The only negative of this experimental set up is that* lacO*-LacI was demonstrated by many groups to block a replication fork itself. Thus on top of the* RTS1*, an additional potentially fork blocking region was inserted to follow the kinetics of HR protein recruitment. It seems that insertion of the* lacO *array has little impact on gene conversion frequencies between ade6 alleles next to* RTS1*; however, replication fork progression was not evaluated by 2D gels. It was somewhat surprising that about 20% of cells with* RTS1-IO *(in inactive orientation) show co-localization of LacI and Rad52. This might suggests that* lacO*-LacI indeed itself is at least a weak fork barrier? You need to clarify whether the* lacO*-LacI interaction does or does not act as RFB. It would seem that the best way to address this question is by 2D gel electrophoresis. Such analysis would also permit measurement of the kinetics of fork stalling at* RTS1-AO*, in relationship to the kinetics of Rad52 foci formation. Therefore, this important issue needs to be resolved by a direct method such as, for example, 2D gel electrophoresis*.

We have done the requested 2D gel analysis of replication intermediates across the *lacO* array under conditions with no LacI and with the same low-level of LacI as in our imaging experiments (see Figure 3—figure supplement 1). We observe no significant difference in the Y-arc with and without LacI, and therefore conclude that the low-level of LacI used in our imaging experiments does not unduly perturb fork progression. It is worth pointing out that among the 20% of *RTS1-IO* cells that exhibit a Rad52 focus that co-localizes with *lacO*-LacI, ∼60% show co-localization in only a single time point, and so far we have never seen more than three time points of co-localization in 90 minutes following anaphase. We suspect that most or all of these co-localizations represent Rad52 recruitment to other problems that will occur stochastically in this region of chromosome 3 (note that events 50 kb on either side of *lacO*-LacI will probably appear to co-localize with it).

*3) The high (unchanged) viability of recombination-defective yeast cells following replication collapse should be measured by using more quantitative approaches. The results from spot tests measuring viability of cells undergoing replication collapse are not sufficiently compelling (in*
Figure 6
*and in the subsection headed “RDR is not always required for cell viability following replication fork blockage at* RTS1*”)*.

We have done the requested quantitative experiment (see Figure 8). Consistent with our spot assay data, we observe no significant difference in viability between *RTS1-IO* and *RTS1-AO* cells.

*4) You must show the “data not shown” referred to in the subsection of the Results entitled “Experimental system”. This is not a request for additional experimentation, but rather to provide the data that you already have. In presenting these unpublished data, you need to explain how you estimated, from*
Figure 1*, the following statement: “It also shows that ∼ 50% of forks remain blocked at* RTS1-AO *long enough for replication to be completed by the opposing fork resulting in fork merging at* RTS1*”*.

The data and quantification are shown in Figure 1—figure supplement 1 of the revised manuscript. The method of quantification is described in part in Materials and methods and in the Figure 1—figure supplement 1 legend.

*5) The following are important clarifications that are required in the manuscript; none of these clarifications require additional experimentation*.

*A)*
Figure 7*. It is unclear why you claim that a replication fork converging with a D-loop would dissociate the D-loop. Why should Mcm2-7 migrating on the leading strand template displace an invading/extended 3' strand that is on lagging strand template*?

We speculate that D-loop dissociation may come from the Pif1 family helicase Pfh1 translocating on the lagging template strand. Pfh1 has been shown to associate with the replisome and we have previously published evidence that it plays a role during replication fork convergence (see [54]). To clarify this we have added the following text to the subsection headed “A hypothetical model for RFB-induced RDR” of our revised manuscript: “This might be advantageous during convergence with a normal replication fork, which could drive dissociation of the D-loop through the action of the replisomes’s accessory DNA helicase Pfh1 translocating on the lagging template strand, thereby converting the restarted “fork” back into a reversed fork at which fork merging could occur (Figure 9)”. In addition, we have included question marks in the diagram (Figure 9) to emphasize the speculative nature of our model.

*B) While discussing your results, you propose that Rad52 is recruited to the position of fork collapse, but not breakage. However, it remains unclear how you can distinguish between these two events in your experimental conditions. For example, it appears that frequent formation of deletions that was frequently observed in this and also in your previous paper (see*
[1]*) could be explained by SSA repair, and therefore induced by DNA breakage. Importantly, approximately a half of such deletions were independent of Rad51 (*[1]*), which is consistent with SSA. In addition, it remains unclear how you explain the increase of the fraction of deletions (which also could be SSA) among secondary recombination events induced by RDR*.

The following text additions have been made in the Discussion:

“Importantly, in wild-type cells breakage of forks blocked at […] DSB-independent single-strand annealing that could occur during fork convergence.”

“The high frequency of ectopic recombination downstream of the RFB (Figure 5) […] to trigger the possibility of re-invasion into *ade6-M375* and the formation of a *ade6*^*+*^ gene conversion.”

*C) You often use “majority”, “minority”, “few”, etc., instead of exact numbers. You need to provide actual values throughout the paper to provide some quantitative context for these relative descriptors. Just a few examples are*:

*i) In the subsection “Live cell imaging of RDR at* RTS1*”: “The majority of cells exhibited a single Rad52 focus” means 51% or 99%. Please provide an exact number*.

*ii) In the subsection “Timing of Rad52 recruitment following replication fork blockage”: “In a few cells…”. Please provide the number instead of “few”. By eye it looks like at least 1/4 of all cells with co-localization*.

*iii) The words “rapid” and “early” in the Title and Abstract are not informative, unless it is clear which quantity they being compared to. I suggest rephrasing*.

The following text changes have been made:

Title: “Recombination occurs within minutes of replication blockage…” (Note that we had to change “impedance” to “blockage” and “leading to” to “producing” to stay within the 120 character limit).

Abstract: “The restarted fork is very susceptible…” has been changed to “The restarted fork is susceptible…”

Abstract: “to warrant the early induction of recombination…” has been changed to “to warrant the induction of recombination…”

Introduction: “We show that Rad52 is recruited to *RTS1* in the majority of cells within minutes of fork blockage and seemingly gives rise to RDR.”

Introduction: “Surprisingly, despite the high frequency of the recombination response.”

Results: “The majority (≥79%) of cells exhibited a single Rad52 focus at varying time points mainly between 20–90 minutes post-anaphase (Figure 3).”

Results: “In a few cells (∼5%) we observed Rad52 foci first co-localizing with *lacO*-LacI at 60–80 minutes post-anaphase (Figure 3).”